# Ligand-induced type II interleukin-4 receptor dimers are sustained by rapid re-association within plasma membrane microcompartments

David Richter[1], Ignacio Moraga[2,3], Hauke Winkelmann[1], Oliver Birkholz[1], Stephan Wilmes[1], Markos Schulte[4], Michael Kraich[4], Hella Kenneweg[1], Oliver Beutel[1], Philipp Selenschik[1], Dirk Paterok[1], Martynas Gavutis[1], Thomas Schmidt[5], K. Christopher Garcia[2,3], Thomas D. Müller[4] & Jacob Piehler[1]

The spatiotemporal organization of cytokine receptors in the plasma membrane is still debated with models ranging from ligand-independent receptor pre-dimerization to ligand-induced receptor dimerization occurring only after receptor uptake into endosomes. Here, we explore the molecular and cellular determinants governing the assembly of the type II interleukin-4 receptor, taking advantage of various agonists binding the receptor subunits with different affinities and rate constants. Quantitative kinetic studies using artificial membranes confirm that receptor dimerization is governed by the two-dimensional ligand–receptor interactions and identify a critical role of the transmembrane domain in receptor dimerization. Single molecule localization microscopy at physiological cell surface expression levels, however, reveals efficient ligand-induced receptor dimerization by all ligands, largely independent of receptor binding affinities, in line with the similar STAT6 activation potencies observed for all IL-4 variants. Detailed spatiotemporal analyses suggest that kinetic trapping of receptor dimers in actin-dependent microcompartments sustains robust receptor dimerization and signalling.

[1] Department of Biology, University of Osnabrück, Barbarastr. 11, 49076 Osnabrück, Germany. [2] Howard Hughes Medical Institute, Stanford University School of Medicine, 279 Campus Drive, Stanford, California 94305-5345, USA. [3] Department of Molecular and Cellular Physiology, and Department of Structural Biology, Stanford University School of Medicine, 279 Campus Drive, Stanford, California 94305-5345, USA. [4] Department of Molecular Plant Physiology and Biophysics, Julius-von-Sachs Institute, University of Würzburg, Julius-von-Sachs Platz 2, 97082 Würzburg, Germany. [5] Physics of Life Processes, Leiden Institute of Physics, Leiden University, Niels Bohrweg 2, 2333 AC Leiden, The Netherlands. Correspondence and requests for materials should be addressed to J.P. (email: piehler@uos.de).

Cytokines are first messenger proteins with key functions in immunity and haematopoiesis. Cytokine signalling therefore has substantial potential for medical intervention and several successful therapies have been approved[1–3]. Cytokines engage their receptors by simultaneously interacting with two or more receptor subunits, which in most cases activates phosphorylation cascades via Janus family tyrosine kinases associated with the cytoplasmic domain of the receptors. The spatiotemporal organization of cytokine receptors in the plasma membrane and in particular the role of receptor dimerization, however, has remained controversially debated[4–6]. While originally receptor dimerization by the ligand has been proposed[7], ligand-independent pre-dimerization of the receptor subunits of homodimeric class I cytokine receptors was observed[8–11]. A similar mechanism was reported to hold true for several heterodimeric class I (refs 12–14) and class II cytokine receptors[15–17]. Other reports suggest that ligand-independent co-clustering of receptor subunits promoted by plasma membrane microcompartmentation may support receptor assembly[18–21]. Indeed, the relatively low, micromolar affinity of many cytokines towards the accessory subunit implies that recruitment of receptor subunits randomly distributed in the plasma membrane could be rather inefficient, in particular because many cytokine receptors are expressed at a level of only a few 100 copies per cell[22]. To avoid artefacts caused by receptor overexpression, we have developed single molecule imaging techniques based on posttranslational labelling via strictly monomeric tags to visualize and quantify receptor dimerization in the plasma membrane of living cells at physiological receptor expression levels[23]. Initial applications of these techniques to class I and class II cytokine receptors were in line with the original model of ligand-induced dimerization[23–25].

Here, we focus on the assembly of the type II interleukin-4 (IL-4) receptor, which is structurally and functionally well characterized[26]. This receptor, which is comprised of the two subunits IL-4Rα and IL-13Rα1, can be activated by IL-4 and by interleukin-13 (IL-13). IL-4 and IL-13 have been associated with allergy, asthma and inhibition of autoimmunity[2], but also seem to be involved in cancer stem cell homeostasis[27] and therefore the receptor is a promising therapeutic target[28,29]. However, for systematic drug development, a quantitative understanding of the molecular and cellular determinants governing the receptor activation at the cell surface is required, which so far remained unresolved[4]: while a recent model proposed that dimerization of IL-4Rα and IL-13Rα1 requires uptake into endosomes[30,31], single molecule studies suggested substantial receptor dimerization by IL-13 at the plasma membrane[25]. IL-4 and IL-13 recognize the receptor subunits IL-4Rα and IL-13Rα1 with differential affinity and kinetics[26]: IL-4 binds to IL-4Rα with sub-nanomolar affinity compared to the micromolar affinity interaction with IL-13Rα1 ($\sim 2.5\,\mu M$)[32], which is $\sim 5$-fold increased in presence of IL-4R-EC (ref. 26). In contrast, IL-13 binds IL-13Rα1 with moderate affinity (30 nM)[26], while no interaction with IL-4Rα is detectable[32]. The IL-13/IL-13Rα1-EC complex, however, binds to IL-4Rα with a binding affinity of 10–20 nM (ref. 33). Recently, IL-4 agonists with strongly altered affinities to IL-13Rα1 have been generated[34]. Assuming a two-step dimerization model, a complex assembly of IL-4Rα and IL-13Rα1 with substantially different efficiency is expected for these different agonists, depending on the affinity to the respective low-affinity subunit (Fig. 1a).

Taking advantage of the versatile intrinsic and engineered features of this system, here we aimed to identify the molecular and cellular determinants governing the assembly and dynamics of the type II IL-4 signalling complex in a quantitative manner. For this purpose, we quantified receptor dimerization in artificial

membranes and in the plasma membrane of living cells. We confirm that the two-dimensional equilibria and stabilities of ternary ligand–receptor complexes reconstituted in artificial membranes correlate with ligand binding affinities and kinetics. Dual-colour single molecule imaging reveals random, uncorrelated distribution of individual IL-4Rα and IL-13Rα1 in the plasma membrane, yet highly robust and efficient ligand-induced dimerization. Interestingly, we find similar dimerization efficiencies for all ligands, despite their large differences in binding affinity. More detailed analysis of the spatiotemporal receptor dynamics suggests that plasma membrane microcompartmentation by the cortical actin cytoskeleton promotes efficient re-association of dissociated receptor dimers, thus ensuring robust maintenance of activated signalling complexes until endocytosis.

## Results

**Dimerization critically depends on receptor densities**. We characterized the molecular features of the type II IL-4 receptor assembly under well-defined conditions *in vitro* using a model system based on solid-supported membranes (SSMs)[35]. For this purpose, the ectodomains of IL-4Rα and IL-13Rα1 fused to a C-terminal decahistidine tag (IL-4Rα − EC and IL-13Rα1-EC, respectively) were tethered onto SSMs by means of a lipid analogue functionalized with tris-NTA[36]. Thus, oriented membrane anchoring and two-dimensional mobility of the receptor subunits in the plane of the membrane was mimicked. Receptor assembly was monitored by simultaneous total internal reflection spectroscopy and reflectance interference (TIRFS-RIf) detection[35]. For ligand binding assays, recombinant IL-4 variants and IL-13 fused to an N-terminal ybbR-tag (11 amino acids) were produced and site-specifically labelled by phosphopantetheinyl-transfer using Coenzyme A conjugated with ATTO 488 or DY647 (ref. 37). All proteins were purified to monodispersity and homogeneity as confirmed by size exclusion chromatography (SEC) (Supplementary Fig. 1) and by SDS-PAGE (Supplementary Fig. 2), respectively. In order to eliminate non-specific binding of IL-4 to the complexed Ni(II) ions, the mutation H59Y outside the receptor binding sites (Supplementary Fig. 3A) was introduced. Neither the ybbR-tag nor the H59Y mutation affected receptor binding properties or activity of IL-4 (Supplementary Fig. 4A,B) and therefore these modifications will not be explicitly declared in the following. Moreover, the mutation K84D (indicated by the subscript D in the following) within the IL-4Rα binding site (Supplementary Fig. 3B) was introduced in order to increase the dissociation rate constant of the IL-4/IL-4Rα complex to a similar level as of the IL-13/IL-13Rα1 complex. This mutation also eliminated mass transport-limited association and strong rebinding during dissociation observed for IL-4 as the association rate constant of the IL-4/IL-4Rα interaction was $\sim 10$-fold reduced (from $1.3 \pm 0.3 \cdot 10^7\,M^{-1}\,s^{-1}$ to $1.7 \pm 0.4 \cdot 10^6\,M^{-1}\,s^{-1}$ (ref. 32)), thus ensuring unbiased analyses of ligand dissociation curves. Importantly, the interaction with IL-13Rα1 is not affected by this mutation (Supplementary Fig. 4C). Therefore, the receptor assembly mechanism depicted in Fig. 1a (left) can still be assumed valid making this mutant suitable for probing the $K_D^T$ of IL-13Rα1 recruitment into the ternary complex. Mutations and tagging as well as labelling of all purified proteins used in this study are summarized in Supplementary Tables 1 and 2.

After quantifying the rate constants of the fluorescently labelled IL-4 variants and IL-13 interacting with the individual receptor subunits by total internal reflection fluorescence spectroscopy (TIRFS) detection (Supplementary Fig. 5 and Supplementary Table 3), receptor dimerization was probed with both subunits

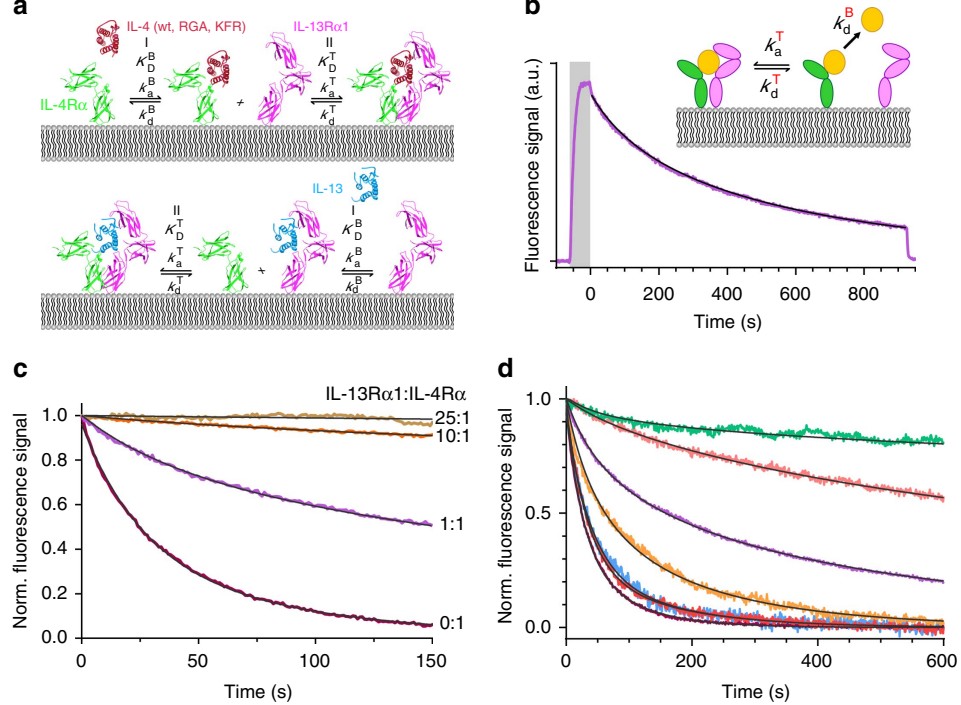

**Figure 1 | Type II IL-4 receptor dynamics and dimerization quantified *in vitro*.** (**a**) Two-step dimerization of the type II IL-4 receptor by different ligands. Top: binding of IL-4 to IL-4Rα followed by recruitment of IL-13Rα1 at the membrane. For IL-4 mutants with altered affinities to IL-13Rα1, changes in the equilibrium between binary and ternary complexes are expected. Bottom: binding of IL-13 to IL-13Rα1 followed by recruitment of IL-4Rα. (**b**) Typical binding assay showing association (highlighted in grey) and dissociation of $^{AT488}$IL-4$_D$ interaction with IL-4Rα-EC and IL-13Rα1-EC tethered on an SSM (TIRFS signal only, full TIRFS-RIf experiment shown in Supplementary Fig. 6). The dissociation curve was fitted (black curve) by using a two-step disassembly model (inset cartoon) to obtain the two-dimensional association rate constant $k_a^T$. (**c**) Comparison of the dissociation kinetics observed for $^{AT488}$IL-4$_D$ interacting with IL-4Rα-EC (1.3 fmol mm$^{-2}$) in absence of IL-13Rα1-EC and in presence of different IL-13Rα1-EC surface concentrations (ratios are indicated at the curves). (**d**) Comparison of the dissociation kinetics obtained for $^{AT488}$KFR$_D$ (green), $^{OG488}$IL-13 (light red), $^{AT488}$IL-4$_D$ (violet), $^{AT488}$RGA$_D$ (orange) and $^{AT488}$DN$_D$ (blue). Binary complex dissociation kinetics of $^{AT488}$IL-4$_D$ ↔ IL-4Rα-EC (dark rose) and $^{OG488}$IL-13 ↔ IL-13Rα1-EC (red) are also depicted.

**Table 1 | 2D kinetic and equilibrium constants of ternary complex formation.**

| Interaction | $k_a^T$ (mol$^{-1}$ mm$^2$ s$^{-1}$)* | $k_d^T$ (s$^{-1}$)* | | $K_D^T$ (molecules µm$^{-2}$) | |
|---|---|---|---|---|---|
| IL-4/IL-4Rα ↔ IL-13Rα1 | $(3.8 \pm 1.0) \cdot 10^{15}$ | $0.23 \pm 0.04$ | $36 \pm 15$* | $1.6 \pm 0.4$† | $0.1 \pm 0.1$‡ |
| IL-13/IL-13Rα1 ↔ IL-4Rα | $(1.1 \pm 0.6) \cdot 10^{14}$ | $0.0023 \pm 0.0007$ | $13 \pm 11$* | n.d. | $0.4 \pm 0.2$‡ |
| KFR/IL-4Rα ↔ IL-13Rα1 | — | $< 0.002$ | $< 5$* | $0.3 \pm 0.1$† | $0.4 \pm 0.2$‡ |
| RGA/IL-4Rα ↔ IL-13Rα1 | — | $> 0.5$ | $296 \pm 104$* | $(> 5)$ | $1.2 \pm 0.6$‡ |

*Molecular interactions determined with receptor ectodomains tethered onto SSMs.
†Molecular interactions determined with transmembrane receptor reconstituted into PSMs.
‡Effective $K_D^T$ estimated from single molecule dimerization assays in the plasma membrane of living cells.

tethered onto an SSM at surface concentrations of ~5 fmol mm$^{-2}$ (~3,000 molecules per µm$^2$) (Fig. 1b and Supplementary Fig. 6). Dissociation of $^{AT488}$IL-4$_D$ was substantially slower compared to experiments with only IL-4Rα-EC (Fig. 1c), which can be explained by recruitment of IL-13Rα1-EC to form a ternary complex. The change in dissociation kinetics was used for quantifying the equilibrium between binary and ternary complexes by fitting a kinetic model schematically depicted in the inset of Fig. 1b (ref. 35). Control experiments at elevated IL-13Rα1-EC surface concentrations revealed substantially further reduced dissociation kinetics (Fig. 1c) in line with a shift of the two-dimensional equilibrium towards ternary complex formation. Moreover, substantially enhanced dissociation of $^{AT488}$IL-4$_D$ was observed upon blocking free IL-13Rα1-EC on the SSM by injection of IL-13 (Supplementary Fig. 7). These results corroborate the formation of a dynamic equilibrium between binary and ternary complexes as depicted in

Fig. 1a that can be quantified from the ligand dissociation kinetics. From the fit of the ligand dissociation kinetics, a two-dimensional (2D) equilibrium dissociation constant $K_D^T$ of $(36 \pm 15)$ molecules per µm$^2$ was obtained for the interaction of IL-4Rα-EC/IL-4$_D$ with IL-13Rα1-EC.

**Ligand binding affinities determine receptor dimerization.** We employed this assay to compare the dimerization by IL-4 mutants with altered 3D binding affinities towards IL-13Rα1, namely RGA (~5-fold increased $K_D$) and KFR (~440-fold decreased $K_D$)[34] as well as DN with no measurable affinity to IL-13Rα1 (ref. 38). These mutants were combined with the H59Y and K84D mutations for probing recruitment of IL-13Rα1-EC. The expected shift in the equilibrium between binary and ternary complexes compared to $^{AT488}$IL-4$_D$ was clearly confirmed by surface binding assays (Fig. 1d and Table 1): for $^{AT488}$RGA$_D$, a

nearly tenfold increase in the $K_D^T$ compared to $^{AT488}$IL-4$_D$ was observed; by contrast, more than 30-fold more efficient ternary complex formation compared to $^{AT488}$IL-4$_D$ was found for $^{AT488}$KFR$_D$. In the latter case, determination of $K_D^T$ by this technique was obstructed because the ligand dissociation kinetics was too slow to be reliably analysed, which is in very good agreement with the $\sim$400-fold increased binding affinity of KFR versus IL-4 for IL-13R$\alpha$1 (ref. 34). By contrast, no significant receptor dimerization was detectable for $^{AT488}$DN$_D$ (Fig. 1d), in line with its dramatically reduced IL-13R$\alpha$1 binding affinity. Analogous TIRFS experiments were carried out using $^{OG488}$IL-13 (Fig. 1d and Supplementary Fig. 5D), for which the opposite sequence of subunit binding is assumed (Fig. 1a, bottom). For the interaction of IL-13R$\alpha$1-EC/IL-13 with IL-4R$\alpha$-EC a $K_D^T$ of $(13 \pm 11)$ molecules per $\mu$m$^2$ was obtained, suggesting a somewhat more efficient ternary complex formation as for IL-4$_D$. Taken together, these *in vitro* experiments confirmed that the molecular 2D dimerization affinities are scaling with 3D binding affinities determined for different agonists.

**2D complex stabilities correlate with binding affinities**. Different binding affinities towards the receptor subunits are accompanied by changes in the interaction kinetics, which has important implications for the formation and the lifetime of individual complexes. We employed chasing experiments in combination with Förster resonance energy transfer (FRET) to quantify the two-dimensional dissociation kinetics of receptor dimers at the membrane[39,40]. For this purpose, IL-13R$\alpha$1-EC site-specifically labelled with Alexa Fluor 568 via an N-terminal ybbR-tag ($^{AF568}$IL-13R$\alpha$1-EC) and IL-4 carrying an acceptor fluorophore ($^{DY647}$IL-4) were used to monitor the dissociation of IL-13R$\alpha$1 from IL-4/IL-4R$\alpha$ in the plane of the membrane as depicted in Fig. 2a. For these experiments, the K84D mutation of IL-4 was not required as the 2D dissociation was probed and ligand dissociation from IL-4R$\alpha$ was not necessary. After formation of the ternary IL-4R$\alpha$-EC/$^{DY647}$IL-4/$^{AF568}$IL-13R$\alpha$1-EC complex at approximately stoichiometric amounts, a large excess of IL-4R$\alpha$-EC ($\sim$10-fold) was tethered onto the membrane. By injecting unlabelled IL-4, which very rapidly and tightly binds to the free IL-4R$\alpha$-EC, exchange of labelled versus unlabelled IL-4 in ternary complex with IL-4R$\alpha$-EC and $^{AF568}$IL-13R$\alpha$1-EC was initiated. The kinetics of this process is limited by the 2D dissociation of $^{DY647}$IL-4/IL-4R$\alpha$ from $^{AF568}$IL-13R$\alpha$1-EC, which was monitored by a decay in FRET efficiency that could be observed in both the donor and the acceptor channel (Fig. 2a, dashed rectangle). Comparison of the mass signal for IL-4 binding to the surface and the recovery of the donor fluorescence confirmed that the 2D dissociation of the ternary complex could be quantified by this method (Fig. 2b), yielding a 2D dissociation rate constant $k_d^T$ of $(0.23 \pm 0.04)$ s$^{-1}$. In case of $^{DY647}$RGA, significantly faster dissociation of the ternary complex was observed, which could not be temporally resolved by this assay (Fig. 2c and Supplementary Fig. 8). In contrast, a more than tenfold decreased $k_d^T$ was estimated for $^{DY647}$KFR (Fig. 2c and Supplementary Fig. 9), in line with its increased IL-13R$\alpha$1 binding affinity. We also quantified the stability of the IL-13R$\alpha$1-EC/IL-13 interaction with IL-4R$\alpha$-EC— the interaction predicted to determine the lifetime of ternary complexes formed by IL-13. Since IL-13 exhibits a tenfold faster dissociation from binary than from ternary complexes, the 2D dissociation rate constant $k_d^T$ could be quantified by fitting the ligand dissociation kinetics of an appropriate ligand chasing experiment (Supplementary Fig. 10)[40]. The $k_d^T$ of $(0.0023 \pm 0.0007)$ s$^{-1}$ obtained for this interaction (IL-13R$\alpha$1-EC/IL-13 $\leftrightarrow$ IL-4R$\alpha$-EC) is similar as the estimated $k_d^T$ of the

IL-4R$\alpha$-EC/KFR $\leftrightarrow$ IL-13R$\alpha$1-EC complex, while the 2D association rate constant is more than 30-fold lower compared to IL-4 (Table 1).

**Enhanced dimerization of the transmembrane receptor**. These measurements yielded a comprehensive quantitative picture of the molecular interactions between the ligand and the receptor ectodomains involved in the assembly of the type II IL-4 receptor on membranes (summarized in Table 1 and Supplementary Fig. 11). The absolute $K_D^T$ values obtained for the different agonists predict substantial differences in IL-4R$\alpha$/IL-13R$\alpha$1 dimerization at the cell surface concentrations of endogenous receptors, which typically are below 1 molecule per $\mu$m$^2$ (ref. 25). However, the contribution of the transmembrane domain in receptor dimerization was neglected by this assay. We therefore quantified, by single molecule fluorescence imaging, the dimerization of transmembrane IL-4R$\alpha$ and IL-13R$\alpha$1 reconstituted into polymer-supported membranes (PSMs) at physiologically representative densities. For this purpose, IL-4R$\alpha$ and IL-13R$\alpha$1 fused to an N-terminal SNAPf-tag[41] and truncated C-terminally after the transmembrane helix were produced in HEK 293T cells. After labelling of the SNAPf-tag by addition of BG-DY547 and BG-DY647, respectively, the receptor subunits ($^{DY547}$IL-4R$\alpha$-TM and $^{DY647}$IL-13R$\alpha$1-TM, respectively) were affinity-purified from the detergent-solubilized membrane fraction and transferred into PSMs as described recently[42]. Dual-colour total internal reflection fluorescence (TIRF) microscopy was applied for imaging diffusion and interaction of $^{DY547}$IL-4R$\alpha$-TM and $^{DY647}$IL-13R$\alpha$1-TM at single molecule level (Supplementary Fig. 12A). Receptor dimerization at different concentrations of IL-4, KFR and RGA, respectively, was quantified by single molecule co-localization and co-tracking analyses (Supplementary Movie 1 and Fig. 3)[42]. These experiments confirmed ligand-induced dimerization of the transmembrane receptor as well as the critical role of the affinity to IL-13R$\alpha$1 in receptor dimerization: while strong dimerization by KFR was achieved, dimerization by RGA was just above the significance threshold (Fig. 3). By contrast, the reduced IL-4R$\alpha$ affinity of IL-4$_D$ only shifted the maximum of the curve to a higher ligand concentration, while the amplitude remained unaffected (Fig. 3). These observations corroborated the relevance of 2D binding affinities for receptor dimerization at physiological receptor densities that cannot be compensated by increasing the ligand concentration. By fitting a comprehensive binding model (Supplementary Fig. 12B) to the concentration-dependent dimerization curves, $K_D^T$ values for the transmembrane receptor were determined. Strikingly, the $K_D^T$ obtained under these conditions were 10–20-fold lower compared to the membrane-tethered receptors (Table 1), suggesting that either the receptor orientation or potential interactions imposed by the transmembrane domains contribute to receptor dimerization.

**Similar mobility and STAT6 activation by all IL-4 agonists**. The molecular properties governing receptor assembly confirmed a broad spectrum of 2D binding affinities and rate constants suitable for a systematic correlation with receptor assembly and activation in living cells. At first, the interaction of IL-4 variants and IL-13 site-specifically labelled with DY647 ($^{DY647}$IL-4 and $^{DY647}$IL-13) with the endogenous cell surface receptor was employed for exploring the spatiotemporal organization of signalling complexes in the plasma membrane under physiological conditions by single molecule imaging techniques. Using PEGylated glass coverslips[23], highly specific ligand binding to the surface of HeLa cells, which were confirmed to express IL-4R$\alpha$ and IL-13R$\alpha$1, but not $\gamma_c$ (Supplementary Fig. 13), was observed

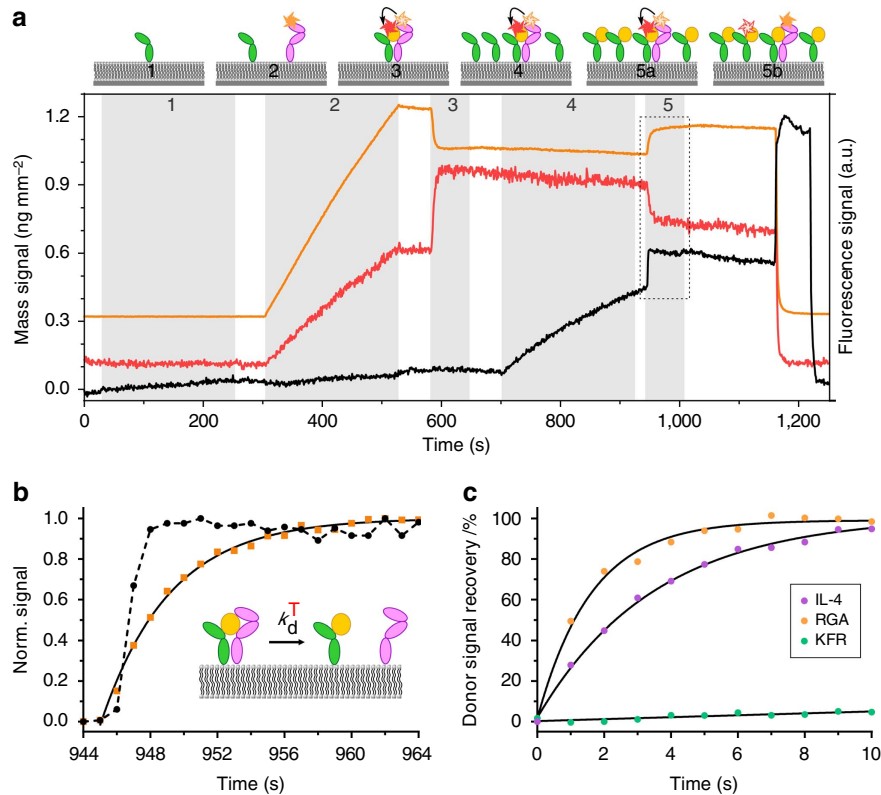

**Figure 2 | Quantification of the ternary complex 2D dissociation kinetics on SSMs by FRET. (a)** Typical assay as monitored by TIRFS-RIf (cartoon depicted at the top): after sequentially tethering (1) IL-4Rα-EC (100 nM) and (2) $^{AF568}$IL-13Rα1-EC (150 nM, FRET donor) onto a SSM, (3) $^{DY647}$IL-4 (100 nM, FRET acceptor) was injected to form a ternary complex in an approximate 1:1:1 stoichiometry. The AF568 fluorescence (orange curve) decreases during $^{DY647}$IL-4 binding due to FRET from $^{AF568}$IL-13Rα1-EC upon ternary complex formation, while in turn the DY647 fluorescence (red curve) increases (dotted rectangle). After washing out unbound ligand, (4) a large amount of IL-4Rα-EC (1 μM) is tethered onto the membrane prior to (5) fast chasing with unlabelled IL-4 (1 μM) that rapidly occupies all excess IL-4Rα-EC on the membrane. Thus, a 2D exchange of $^{DY647}$IL-4/IL-4Rα-EC bound to $^{AF568}$IL-13Rα1-EC by IL-4/IL-4Rα-EC is initiated, which is accompanied by recovery of the donor fluorescence with a rate constant corresponding to $k_d^T$. RIf signal is shown in black. **(b)** The 2D dissociation rate constant $k_d^T$ was obtained by fitting an exponential function (black line) to the AF568 signal (orange points) within the time window highlighted in **a**. The RIf signal of IL-4 binding to excess IL-4Rα-EC (black points and dashed line) is shown to indicate the time resolution of this assay. **(c)** Comparison of the donor fluorescence recovery (that is, dissociation kinetics) obtained for $^{DY647}$RGA (orange), $^{DY647}$IL-4 (violet) and $^{DY647}$KFR (green). Depicted curves show representative single experiments.

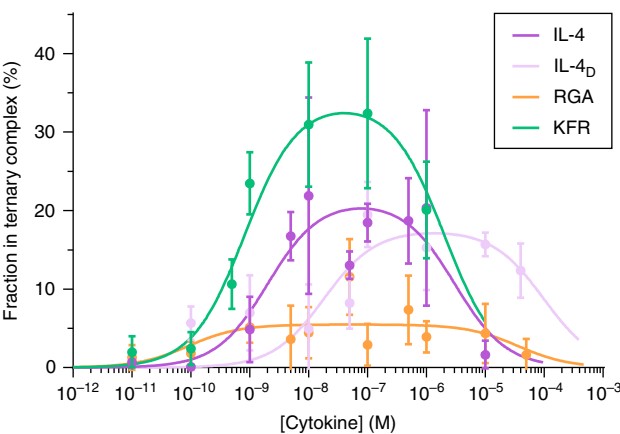

**Figure 3 | Dimerization of the transmembrane receptor reconstituted into PSM.** Single molecule co-localization of $^{DY547}$IL-4Rα-TM and $^{DY647}$IL-13Rα1-TM reconstituted into PSMs at different concentrations of IL-4, IL-4$_D$, KFR and RGA, respectively (mean ± s.d. of 3–5 measurements). The lines are best fits of a comprehensive equilibrium model (cf. Supplementary Fig. 12B).

by TIRF microscopy at single molecule level (Supplementary Movie 2 and Fig. 4a,b). Individual ligands diffusing in the plasma membrane could be discerned as confirmed by single step photobleaching at elevated laser power. Under saturating ligand concentrations (2 nM), similar densities of receptor complexes (0.06–0.08 μm$^{-2}$) were observed for $^{DY647}$IL-4 and its variants $^{DY647}$KFR, $^{DY647}$RGA and $^{DY647}$DN as well as for $^{DY647}$IL-13 (Fig. 4b). No tendency of clustering was observed, but stochastic distribution and random diffusion in the plasma membrane. Under these conditions, individual ligands could be readily tracked with high fidelity (Supplementary Movie 2 and Fig. 4a) and their diffusion properties could be quantified by evaluating the trajectories. Mean square displacement analysis confirmed random diffusion of signalling complexes in the plasma membrane (Supplementary Fig. 14A). Typically, ∼25% of the tracked molecules were immobile, which can in part be ascribed to non-specific adsorption to the coverslip surface as well as to complexes involved in endocytosis. For this reason, the immobile fraction was removed from the data set before further analyses to ensure that only ligands bound to the cell surface were considered. Diffusion constants obtained from step-length histograms (Fig. 4c,d; Table 2 and Supplementary Fig. 14B) revealed a significantly reduced mobility of $^{DY647}$KFR compared to $^{DY647}$DN (Table 2 and Supplementary Fig. 14C). Since DN

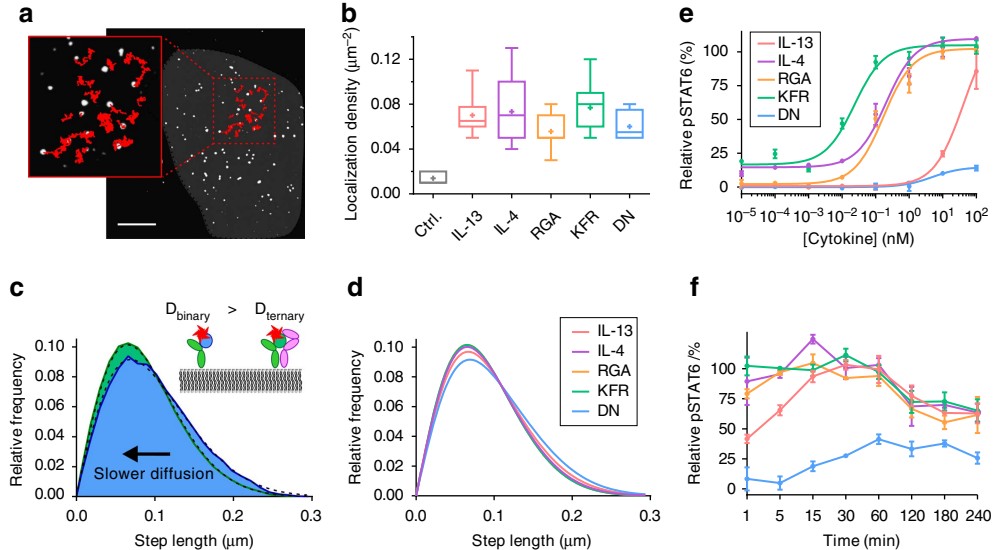

**Figure 4 | Cellular binding and activity of different IL-4 agonists and IL-13.** (**a**) Live-cell single molecule localization and tracking of [DY647]KFR bound to endogenous cell surface IL-4Rα and IL-13Rα1 of a HeLa cell. The underlying cell is indicated by the grey background; single molecule trajectories are depicted as red lines (see also Supplementary Movie 2). Scale bar: 10 μm. (**b**) Cell surface density of bound IL-4 variants and IL-13 quantified by single molecule localization. As negative control, binding of [DY647]IL-4 after preincubation with 20 nM unlabelled KFR was quantified. ($n \geq 10$ cells for IL-13, IL-4, RGA and KFR, $n = 5$ cells for DN and ctrl.) (**c**) Diffusion properties of cell-bound [DY647]DN (blue) and [DY647]KFR (green) presented as step-length distributions, which were fitted (dashed lines) by considering two components corresponding to a slow and a fast mobile fraction. Shift to lower mobility by KFR indicates ternary complex formation in contrast to DN, which does not interact with IL-13Rα1. (**d**) Comparison of the step-length histograms for [DY647]IL-13, [DY647]IL-4, [DY647]RGA, [DY647]KFR and [DY647]DN. Fitted curves in **c,d** are based on data from $n > 350$ single-molecule trajectories ($n = 200$ for DN) with minimum length of 150 frames. Individual step lengths were determined for a time lapse of 32 ms (1 frame) and histogramed as depicted in Supplementary Fig. 14B. (**e,f**) STAT6 phosphorylation activity of different agonists in HeLa cells analysed by phospho-flow cytometry. (**e**) Dose–response curves of STAT6 phosphorylation observed after stimulation for 15 min. (**f**) Kinetics of STAT6 phosphorylation after stimulation with 1 μM of agonist. Data points in **e,f** represent mean ± s.d. of three independent experiments.

**Table 2 | Average diffusion constants obtained from single molecule tracking data (mean ± s.d.).**

| Ligand | D (μm² s⁻¹) |
|---|---|
| DN (primarily binary complex) | 0.113 ± 0.002* |
| KFR (primarily ternary complex) | 0.086 ± 0.001* |
| IL-4 | 0.091 ± 0.001* |
| RGA | 0.088 ± 0.001* |
| IL-13 | 0.099 ± 0.001* |
| IL-4Rα (w/o ligand) | 0.119 ± 0.010† |
| IL-4Rα (w/ KFR) | 0.107 ± 0.002† |
| IL-13Rα1 (w/o ligand) | 0.140 ± 0.012† |
| IL-13Rα1 (w/KFR) | 0.108 ± 0.001† |
| IL-4Rα/IL-13Rα1 dimers (ligand induced) | 0.087 ± 0.003‡ |

*Obtained with DY647-labelled ligands bound to the endogenous receptor.
†Mean values determined in cells co-expressing IL-4Rα and IL-13Rα1 without ligand and after addition of KFR.
‡Mean values from [DY647]IL-4Rα/[TMR]IL-13Rα1 co-trajectories in presence of either IL-4, IL-13 or KFR.

supposedly interacts with IL-4Rα only, these results suggest that a loss in receptor mobility is caused by ligand-induced receptor dimerization. Similar changes in the diffusion properties have been observed for ligand-induced dimerization of the type I interferon receptor[23]. Interestingly, comparable mobility as for KFR was observed for all IL-4 variants (except DN) and IL-13 (Table 2), indicating a similar ability to form signalling complexes in the plasma membrane. In line with these experiments, ligand-induced STAT6 phosphorylation examined by phospho-flow cytometry showed largely overlapping potencies (Fig. 4e) and similar kinetics (Fig. 4f) of STAT6 activation for all IL-4 variants, with only a minor increase in potency observed for KFR. In contrast, significantly lower potency of IL-13 and slower

signalling kinetics compared to the IL-4 variants were observed, which could be related to the substantially lower on-rate that has been shown to be an important determinant for signalling potency[25]. These results suggest efficient receptor assembly by different IL-4 variants irrespective of the large differences in IL-13Rα1 binding affinity, hinting towards a spatiotemporal organization of IL-4Rα and IL-13Rα1 in the cellular context.

**Randomly diffusing receptor subunits are dimerized by IL-4.**
To identify the cellular determinants of receptor dimerization we visualized the receptor subunits in the plasma membrane by dual colour single molecule fluorescence imaging for spatiotemporal correlation of both receptor subunits (Supplementary Fig. 15A–C)[25]. For this purpose, IL-4Rα fused to the SNAPf-tag and IL-13Rα1 fused to the HaloTag[43] were transiently co-expressed in HeLa cells. These tags are engineered enzymes that irreversibly react with their substrates, which is exploited for orthogonal covalent labelling with photostable synthetic fluorescent dyes. Posttranslational labelling was carried out prior to imaging experiments by addition of the substrates BG-DY647 ([DY647]IL-4Rα) and HTL-TMR ([TMR]IL-13Rα1), respectively (Fig. 5a). After washing out excess dye, individual [DY647]IL-4Rα and [TMR]IL-13Rα1 could be discerned at a ~5–10-fold elevated density compared to the endogenous receptor (0.1–0.5 μm⁻², Supplementary Fig. 15D). Single molecule tracking confirmed random diffusion of both receptor subunits (Supplementary Fig. 14A) with similar diffusion properties as observed for DN bound to the endogenous receptor (Table 2). Likewise, a similar level of typically 25% immobile molecules was observed which was excluded from further analyses. Receptor dimerization was analysed by single molecule co-localization/co-

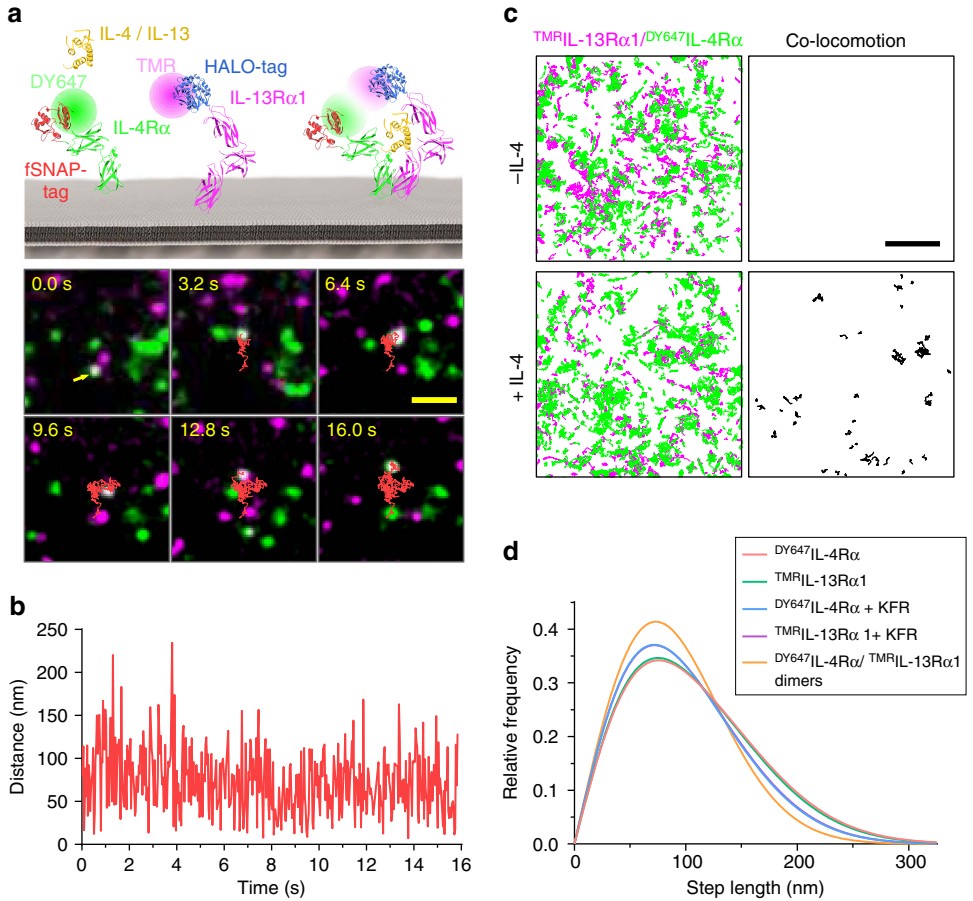

**Figure 5 | Receptor dimerization probed by single molecule co-tracking analysis.** (**a**) Co-tracking assay: IL-4Rα (green) was fused to the SNAPf-tag and labelled with BG-DY647 while IL-13Rα1 (magenta) was fused to the HaloTag and labelled with HTL-TMR. Individual co-localized $^{DY647}$IL-4Rα and $^{TMR}$IL-13Rα1 (white signal highlighted by the arrow) in presence of IL-4 was co-tracked over 500 frames (16 s, cf. Supplementary Movie 3). The corresponding trajectory is depicted as a red line. Scale bar: 2 μm. (**b**) Distance between the two molecules in each frame. (**c**) Co-tracking analysis of $^{DY647}$IL-4Rα and $^{TMR}$IL-13Rα1 in the absence and in the presence of IL-4. Scale bar: 5 μm. (**d**) Diffusion properties of $^{DY647}$IL-4Rα and $^{TMR}$IL-13Rα1 before and after stimulation with 200 nM KFR as well as of ligand induced $^{DY647}$IL-4Rα/$^{TMR}$IL-13Rα1 dimers presented as fitted step-length distributions (time lapse 32 ms, 1 frame), based on two components corresponding to a slow and a fast mobile fraction. Depicted curves are based on data of a representative single cell experiment comprising $n > 400$ trajectories ($n = 24$ for dimers) with min. length of ten frames.

tracking[23] (Supplementary Fig. 15B and Fig. 5a,b). Co-tracking of $^{DY647}$IL-4Rα and $^{TMR}$IL-13Rα1 in absence of ligand did not yield any indication of receptor pre-dimerization (Fig. 5c). Upon addition of IL-4, however, individual $^{DY647}$IL-4Rα/$^{TMR}$IL-13Rα1 dimers diffusing in the plasma membrane could be clearly identified and tracked for several seconds (Supplementary Movie 3 and Fig. 5a–c). Such dimers could be observed for all ligands, yielding similar diffusion properties (Table 2). A small, yet significant reduction in mobility of $^{DY647}$IL-4Rα and $^{TMR}$IL-13Rα1 upon ligand-induced dimerization was noticed (Fig. 5d and Table 2), yielding a diffusion constant matching the diffusion constant obtained for $^{DY647}$KFR bound to endogenous receptor (Table 2). These results clearly confirmed the hypothesis of ligand-induced receptor dimerization in the plasma membrane and suggest a surprisingly high dimerization efficiency for all ligands.

**Efficient receptor dimerization by all ligands.** To quantify the spatial receptor organization and dimerization in the plasma membrane more precisely, we employed particle image cross-correlation (PICCS) analysis[44] at the single molecule level. PICCS directly quantifies the spatial correlation of two spectrally

separated molecule species via a cross-correlation function $C_{cum}$ that was calculated from all molecule positions of a single cell as schematically depicted in the inset of Fig. 6a and in Supplementary Fig. 15C. By fitting the linear component of $C_{cum}$ with equation (1), the correlated fraction α as a measure for non-random co-localization is obtained from the intercept. In absence of ligand, individual $^{DY647}$IL-4Rα and $^{TMR}$IL-13Rα1 molecules did not show any significant α (Fig. 6a), confirming that the receptor subunits are neither pre-dimerized nor pre-organized in the plasma membrane. Instead, the entirely linear shape of the $C_{cum}$ indicates a random receptor distribution in the plasma membrane. After incubating IL-4 at a concentration saturating all IL-4Rα subunits, dimerization could clearly be identified as an increase of α to ∼11% (Fig. 6b,c). For converting α into absolute dimerization levels, a reliable determination of the degree of receptor labelling is required, which was obstructed by the substantial receptor turnover after cell surface labelling, probably caused by rapid, ligand-independent endocytosis of the IL-4 receptor[45]. Taking advantage of the highly asymmetric binding affinities of IL-4 towards the receptor subunits, we therefore used labelled IL-4 to directly quantify the fraction of IL-4Rα-bound IL-4 forming a ternary complex with IL-13Rα1. For this purpose, the interaction of $^{DY647}$IL-4 with $^{TMR}$IL-4Rα

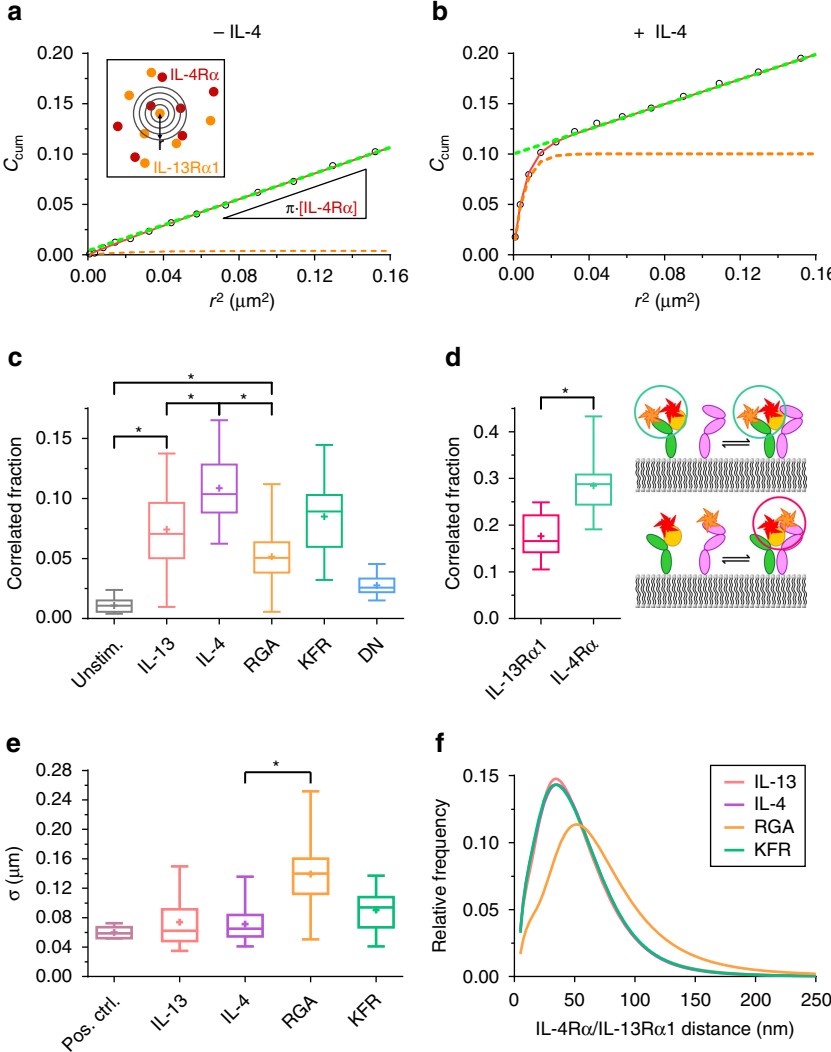

**Figure 6 | Quantification of ternary complex assembly in living cells.** (**a**,**b**) Spatial correlation of $^{TMR}$IL-13Rα1 and $^{DY647}$IL-4Rα molecules for a representative single cell in absence (**a**) and presence (**b**) of IL-4, analysed by PICCS. Black circles: cumulative correlation function $C_{cum}$ obtained from single molecule localizations, averaged over frames 1–20. Dashed green line: linear contribution of the cumulative correlation function (equation (1)). Orange dashed line: cumulative correlation function after subtraction of the linear term from the fitted function (equation (2), red curve). (**c**) Correlated fraction of $^{TMR}$IL-13Rα1 and $^{DY647}$IL-4Rα for the examined ligands, determined by PICCS. (**d**) Quantification of receptor dimerization by PICCS analysis of $^{DY647}$IL-4 bound to $^{TMR}$IL-4Rα and to $^{TMR}$IL-13Rα1, respectively. Data are based on $n = 16$ cells (IL-13Rα1 + IL-4) and $n = 9$ cells (IL-4Rα + IL-4). (**e**) The correlation length (σ) obtained from PICCS analysis. The data shown in **c**,**e** are pooled from at least two independent experiments with $n > 20$ cells ($n = 9$ cells for unstim. and $n = 6$ cells for DN and pos. ctrl.). (**f**) Receptor distance distributions of co-trajectories, induced by KFR, IL-4, IL-13 and RGA, fitted with a three-component log-normal function. Depicted curves are based on data from different cells comprising $n > 50$ trajectories with minimum length of 30 frames. Statistical analysis by a two-sample Kolmogorov–Smirnov test (*$P < 0.05$).

and with $^{TMR}$IL-13Rα1, respectively, was quantified by PICCS in separate experiments (Fig. 6d). A cross-correlated fraction of ~18% was observed for the $^{DY647}$IL-4/$^{TMR}$IL-13Rα1 interaction compared to ~28% for the $^{DY647}$IL-4/$^{TMR}$IL-4Rα complex. These results implicate that 60–70% of IL-4Rα bound to IL-4 on the cell surface forms a ternary complex with IL-13Rα1. Taking into account the average cell surface concentrations of $^{DY647}$IL-4Rα and $^{TMR}$IL-13Rα1 quantified by single molecule localization and corrected for the degree of labelling, an effective $K_D^T$ of ~0.1 molecules per μm$^2$ was estimated for IL-13Rα1 interacting with the IL-4Rα/IL-4 complex. Thus, receptor dimerization in the plasma membrane is >10-fold more efficient compared to the molecular interaction in artificial membranes.

Only minor differences in the dimerization efficiencies were observed for the IL-4 mutants as well as for IL-13 (Fig. 6c, Supplementary Table 4), despite the very large differences in the

molecular interactions observed *in vitro*. While the increase in IL-13Rα1 binding affinity of the KFR mutant did not yield any increase in dimerization efficiency, only a small, yet significant drop in dimerization efficiency was observed for RGA. Even for the variant DN with immeasurable binding affinity to IL-13Rα1, receptor dimerization above the background could be detected, in line with its residual STAT6 phosphorylation activity (Fig. 4e,f). In contrast, the dimerization efficiency of IL-13 turned out to be lower compared to IL-4, despite the overall higher $K_D^T$ of IL-4 observed *in vitro*. The apparently reduced dimerization efficiency of IL-13, however, could be explained by the ~20% excess of IL-13Rα1 used in these experiments. Taken together, these quantitative dimerization assays substantiate that ligand-induced dimerization of IL-4Rα and IL-13Rα1 in the plasma membrane is not exclusively controlled by the molecular binding affinities.

**Increased inter-subunit distance for weak 2D interactions.**
Plasma membrane microcompartmentation plays an intricate role in the spatiotemporal organization of signalling complexes[46–51] and has been proposed to support receptor dimerization and activation by increasing the local concentration of interaction partners or by stabilizing signalling complexes[46,48]. Since PICCS analysis excluded pre-organization of the receptor subunits in the absence of ligand we explored in more detail the spatiotemporal dynamics of receptor dimers after ligand stimulation. For this purpose, we compared, for the different ligands, the correlation length $\sigma$ obtained from fitting $C_{cum}$ with equation (2), which corresponds to the average distance of spatially correlated receptor subunits (Fig. 6e and Supplementary Table 4). Assuming that only $^{DY647}$IL-4R$\alpha$ and $^{TMR}$IL-13R$\alpha$1 dimerized by the ligand contribute to $\alpha$, a theoretical $\sigma$ of $\sim 10$ nm is expected based on the distance in the crystal structure of the ternary complex[26]. The experimental $\sigma$ of $\sim 60$ nm observed in a positive control protein with both SNAPf-tag and HaloTag fused to a transmembrane domain (Fig. 6e) can be ascribed to the limited overall precision of single molecule co-localization. This not only includes the photon-based localization precision and the precision of channel alignment, but also blurring due to the movement of the proteins during imaging. While $^{DY647}$IL-4R$\alpha$ and $^{TMR}$IL-13R$\alpha$1 dimers induced by IL-4, KFR and IL-13 show $\sigma$ values similar to the positive control (70–90 nm), a significantly increased $\sigma$ (140 nm) was observed in case of RGA. The increased average distance of $^{DY647}$IL-4R$\alpha$ and $^{TMR}$IL-13R$\alpha$1 in complex with RGA was accompanied by an increased bandwidth of $\sigma$. This finding was corroborated by directly quantifying the distances in individual $^{DY647}$IL-4R$\alpha$/ $^{TMR}$IL-13R$\alpha$1 dimers identified by dual colour single molecule co-tracking analysis (Fig. 6f). Compared to all other ligands, a substantial shift to higher distances was observed for RGA-induced dimers including a significant fraction of distances above the

co-localization precision (Fig. 6f, Supplementary Fig. 16 and Supplementary Table 5).

**Dimer lifetime depends on the cortical actin skeleton.** We hypothesized that this increased inter-subunit distance observed for RGA was caused by transient 2D dissociation from IL-13R$\alpha$1 followed by rapid re-association due to trapping within sub-microscopic transient confinement zones (TCZ)[46,47,52]. Assuming TCZ dimensions $<100$ nm as reported for diffusion in the plasma membrane[46], such transient dissociation events could not be resolved by the spatiotemporal resolution of the experimental setup used for single molecule co-tracking and therefore apparently enhanced complex stability would be expected. We therefore quantified the lifetime of individual receptor dimers by dual colour single molecule tracking. To directly probe the low-affinity interaction of IL-4R$\alpha$/IL-4 with IL-13R$\alpha$1, we expressed, by means of an inducible promoter[53], HaloTag-IL-13R$\alpha$1 at physiological levels in HeLa cells. After labelling of HaloTag-IL-13R$\alpha$1 with HTL-TMR, $^{DY647}$KFR or $^{DY647}$RGA was added at a concentration sufficient to rapidly bind and saturate endogenous IL-4R$\alpha$. Because *in vitro* labelling of IL-4 is achieved with a yield $>90\%$, dimerized receptor subunits could be well detected at these low concentrations, while co-tracking of $^{DY647}$KFR or $^{DY647}$RGA and $^{TMR}$IL-13R$\alpha$1 was possible with high fidelity. The distribution of co-trajectory lengths' was taken as a semi-quantitative measure for the complex stability. Very similar patterns were obtained for KFR and RGA (Fig. 7a, left), despite the more than 100-fold faster dissociation of RGA from IL-13R$\alpha$1 observed on SSMs. While the experimental trajectory length is limited by photobleaching and tracking fidelity, a significant lower complex stability ($<1$ s) of receptor dimers induced by RGA is expected from the molecular dissociation kinetics.

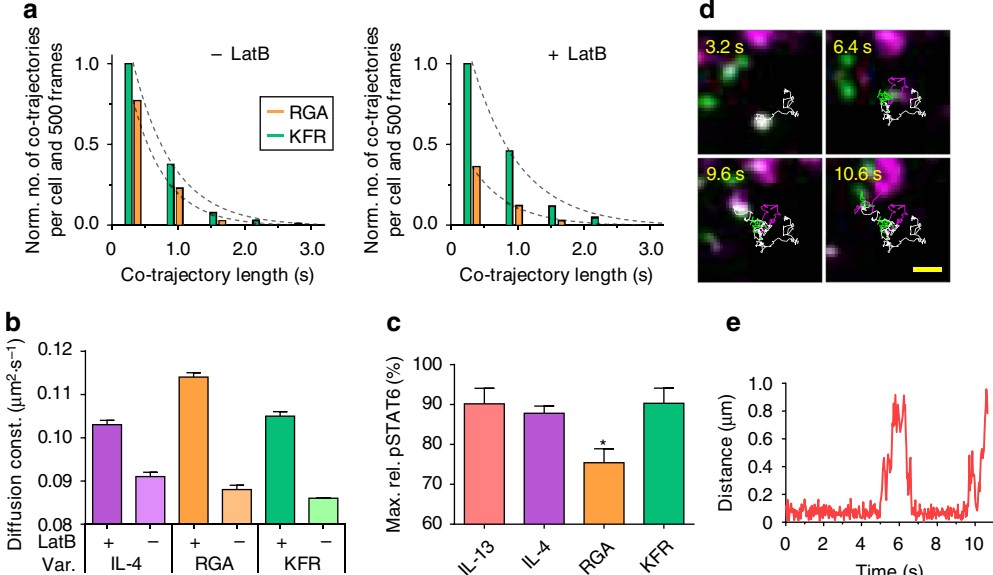

**Figure 7 | Dynamic complex stabilization by the actin cytoskeleton.** (**a**) Co-trajectory length of RGA-induced ternary complexes is dramatically decreased due to disruption of the actin cytoskeleton by LatB (Latrunculin B) treatment. Data are based on $n>60$ ($-$ LatB) and $n>17$ ($+$ LatB) cells. Co-trajectories with less than 30 frames were not considered. (**b**) Diffusion constants of fluorescent ligand-bound receptors in absence and in presence of LatB (mean $\pm$ s.d. of $n>10$ cells). (**c**) Maximum relative pSTAT6 level in presence of LatB is distinctly lowered for RGA. HeLa cells were treated with 10 $\mu$m LatB, stimulated for 15 min with saturating doses of the respective ligand and the relative levels of pSTAT6 compared to untreated cells were analysed by phospho-flow cytometry using anti-pSTAT6 specific antibodies coupled to fluorescent dyes. Error bars correspond to the s.d. from three independent experiments. Statistical analysis by Student's $t$-test ($^*P<0.05$). (**d**) $^{DY647}$IL-4R$\alpha$ (green) and $^{TMR}$IL-13R$\alpha$1 (magenta) complex re-association at single molecule level in presence of RGA. Images showing the complex at different time points of a 330-frame (10.6 s) time-lapse experiment (cf. Supplementary Movie 4) including its corresponding co-trajectory (white). Scale bar: 1 $\mu$m. (**e**) Distance between the two molecules throughout the entire trajectory.

These results supported our hypothesis of complex re-association in submicroscopic TCZ, which were probably caused by the cortical membrane skeleton (MSK)[46]. The MSK of HeLa cells has been reported to have an average mesh size of ~70 nm with a typical residence time of ~5 ms (refs 54,55). To explore the role of the MSK in receptor diffusion and interaction we applied Latrunculin B (LatB), a compound that partially inhibits actin polymerization by binding to monomeric G-actin[56] and thus decreases confinement by the MSK[55,57]. For a minimum bias of other cellular functions, a concentration of LatB was applied (10 μM), which was optimized to dissipate the fine cortical actin structure while major stress fibres and the cell morphology was maintained (Supplementary Fig. 17). Under these conditions, the diffusion constants of IL-4Rα and IL-13Rα1 increased (Fig. 7b), as expected for MSK-limited hop diffusion in the plasma membrane[55,58,59]. By contrast, no significant changes in the diffusion properties of the receptor could be observed upon treatment with cholesterol oxidase (Supplementary Fig. 18), which has been shown to affect the mobility of lipid raft-associated proteins[60]. These results established that microcompartmentation by the MSK is the major determinant of type II IL-4 receptor spatiotemporal dynamics in the plasma membrane.

We therefore explored the contribution of the MSK that has been recently proposed to promote re-association of receptor dimers[47,49,52]. Strikingly, LatB treatment substantially reduced the length of co-trajectories of receptor dimers induced by RGA, but not by KFR (Fig. 7a, right). The mobility of RGA- and KFR-induced dimers, however, was identical after LatB treatment (Supplementary Fig. 19). The importance of MSK-dependent receptor confinement was further supported by the differences in overall mobility observed for IL-4 and mutants in presence of LatB (Fig. 7b): the largest increase was observed for RGA, in line with its higher propensity to exist bound to IL-4Rα only. These results sustained our hypothesis that submicroscopic TCZ based on the MSK support receptor dimerization by RGA.

To investigate the functional relevance of this feature for downstream signalling we quantified the maximum level of STAT6 phosphorylation induced by different agonists in presence of LatB using phospho-flow cytometry. While all ligands showed reduced pSTAT6 activation in presence of LatB, this effect was most pronounced for RGA (Fig. 7c). These results established that the low molecular affinity of RGA can be at least partially compensated by ensuring efficient re-association in MSK-based TCZ.

**A simple spatial-stochastic model mimics receptor dynamics.** These results confirmed that MSK-dependent TCZ were involved in dynamically stabilizing transient receptor dimers, probably by ensuring efficient re-association of dissociated complexes. For some trajectories, dissociation of receptor dimers and re-association of the same subunits could even be followed with the spatiotemporal resolution of our setup (Supplementary Movie 4 and Fig. 7d,e), corroborating the relevance of this process under physiological conditions. To validate this conclusion, we established a simple spatial diffusion-reaction model including a submicroscopic, semipermeable meshwork (Supplementary Movie 5 and Supplementary Fig. 20) as proposed by the MSK concept[54]. The molecular dissociation probabilities and local diffusion properties were parameterized as quantified *in vitro*. By adjusting the hopping and association probabilities, diffusion properties of the receptor subunits and the ternary complex as well as total receptor levels in complex could be closely matched with the experimentally obtained data (Supplementary Movie 6 and Supplementary Fig. 21A–E). Under these conditions, a slightly increased probability (+10%) of molecule collision compared to a fully homogenous diffusion scenario was observed (data not shown), which is caused by the meshwork constraints. The number of re-associations after dissociation of receptor dimers, however, is increased by 125% due to the membrane confinement and the re-association frequency is, as expected, explicitly higher (~4-fold) for less stable complexes

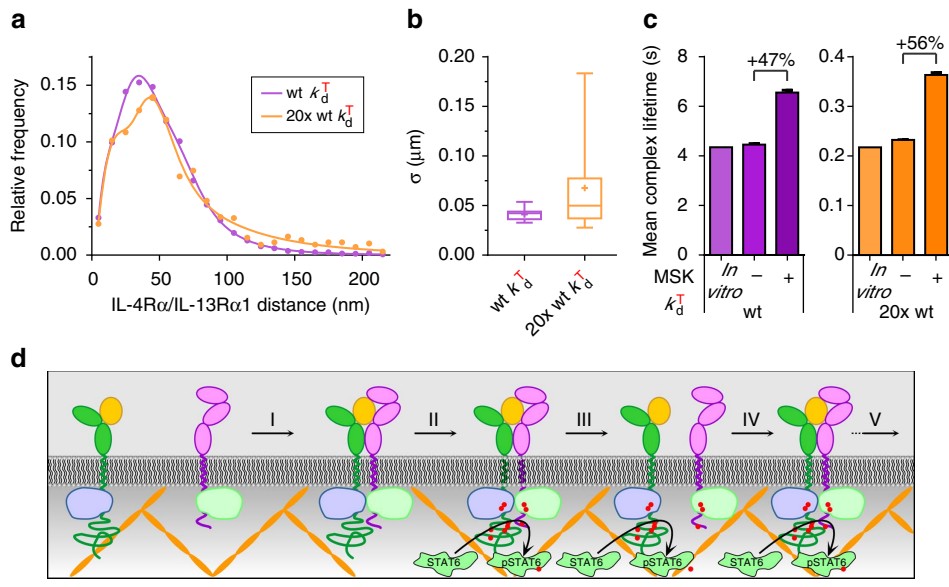

**Figure 8 | Simulation of spatiotemporal receptor dynamics including MSK-dependent membrane compartmentation and hop-diffusion.** (**a**) Representative receptor distance distributions of co-trajectories as well as (**b**) the corresponding correlation lengths, simulated on the basis of the depicted model depicted in Supplementary Fig. 20 with two different ternary complex dissociation constants ($n = 5$) are comparable to the experimentally derived data and verify the adopted model. (**c**) The mean ternary complex lifetime is increased by ~50% in simulation runs including confinement by the MSK (mean ± s.d., $n = 10$). (**d**) Mechanistic model for ligand-induced type II interleukin-4 receptor dimerization sustained by rapid re-association within MSK-dependent TCZ. After two-step receptor assembly of the ternary complex (I), downstream signalling is initiated (II) that is maintained during transient dissociation events (III, IV) until receptor uptake by endocytosis (V).

(Supplementary Fig. 21F). Importantly, the receptor distance distributions of co-trajectories as well as the corresponding correlation lengths extracted from simulation runs with different ternary complex dissociation constants are comparable to the experimentally derived data and reproduce the shift to longer distances/correlation lengths and broader distribution for a low-affinity ligand (like RGA) (Fig. 8a,b) as experimentally observed (Fig. 6e,f). Finally, similar to the negative effect of LatB on the length of RGA-induced co-trajectories (Fig. 7a), our model simulations revealed a ∼50% increased mean ternary complex lifetime (measured as the time both receptor subunits reside in the same compartment after undergoing at least one association event) when comparing simulations with and without constrained diffusion by the MSK (Fig. 8c and Supplementary Fig. 21G,H). This effect was essentially independent on the molecular complex lifetime (Fig. 8c), highlighting that the re-association probability depends on the 2D association kinetics[52], which was assumed identical for both cases. Owing to the relatively short observation times and limiting co-tracking fidelity, the effect was experimentally most prominent for the short-lived dimers formed by RGA.

## Discussion

With its critical implication in allergy, asthma and cancer, the type II IL-4 receptor is an important drug target[28,29]. For efficient pharmaceutical targeting of receptor activation, detailed understanding of its mode of engagement by the ligand is desired, which is, however, controversially debated. Focusing on the early events of ligand interaction with the receptor subunits in the plasma membrane, we have here untangled the molecular and cellular determinants governing assembly of the type II IL-4 receptor signalling complex. The 2D interactions involved in receptor dimerization at the membrane were quantified by *in vitro* assays with the ectodomains tethered onto SSMs as well as with the transmembrane receptor reconstituted into PSMs. These measurements confirmed ligand-induced receptor dimerization and very different abilities of IL-4 mutants to recruit IL-13Rα1 into the ternary complex. The $K_D^T$ and $k_d^T$ values of the 2D ligand–receptor interactions limiting receptor dimerization were in line with the relative binding affinities of the different ligands. Yet, all ligands showed substantially enhanced dimerization of the transmembrane receptor subunits in PSMs compared to the $K_D^T$ determined on SSMs, suggesting that the transmembrane helix either structurally organizes the receptor subunits in a favourable orientation on the membrane or directly mediates contacts stabilizing the complex. The molecular basis of this striking contribution of the transmembrane domain for ligand-induced receptor dimerization needs to be investigated in more detail in the future.

Although strong differences in dimerization of transmembrane IL-4Rα and IL-13Rα1 in PSMs was observed for different IL-4 variants, their functional properties in living cells in terms of receptor binding, diffusion and early signal activation proved to be very similar, suggesting that spatial co-organization may promote dimerization in the plasma membrane. Detailed single molecule analyses, however, revealed random and uncorrelated distribution of IL-4Rα and IL-13Rα1 in the plasma membrane, yet very efficient dimerization by all ligands. The high degree of ligand-induced receptor dimerization revealed by single molecule imaging at physiological cell surface expression levels was in agreement with previous estimations for the related type I IL-4 receptor based on competitive antibody binding experiments[61]. By contrast, recent live cell fluorescence cross-correlation spectroscopy studies[30] suggested much less efficient dimerization of the type II IL-4 receptor in the plasma

membrane. Importantly, substantially higher receptor densities were required for these measurements (>1,000-fold above the endogenous cell surface expression level), which may account for these fundamental differences. Indeed, our results indicate that the cellular context may play an important role for ensuring efficient receptor assembly, for example, by enhancing the local subunit concentrations by partitioning into membrane microdomains. However, co-organization or co-clustering of receptor subunits in the absence of ligand, which has been proposed for several cytokine receptors[18–21], was excluded by spatial correlation techniques. Receptor dimers did not exhibit any tendency of partitioning at the plasma membrane either, but diffused freely while showing only a minor loss of mobility upon dimerization, which can be explained by the increased friction upon assembly of receptor dimers. These observations support that ligand-induced dimerization of receptor subunits randomly distributed in the plasma membrane is the key event for activation of cytokine receptors.

Detailed studies of the spatiotemporal dynamics of receptor dimers revealed a critical role of MSK-based TCZ in dynamically stabilizing transient receptor dimers. Such re-association events have been predicted to enhance signalling by promoting signalling bursts[46,47] and recent fast quantum dot tracking has supported their relevance for the kinetics of receptor dimerization[49,52]. By a simple spatial-stochastic model fully parameterized by experimentally derived constants, we could reproduce essential features of the spatiotemporal receptor dynamics in the plasma membrane. This model corroborates the hypothesis that passive confinement by the MSK ensures rapid re-association of transient receptor dimers. While cholesterol depletion experiments suggest that lipid-based segregation does not significantly contribute to receptor dimerization, MSK-dependent hierarchical TCZ as observed by recent single particle tracking studies with very high spatiotemporal resolution[49,52] may further enhance this effect and thus explain the residual discrepancies between experimental and simulated data. However, rather than shifting the equilibrium by local enrichment of the interaction partners, MSK-based TCZs ensure rapid re-association of the same receptor subunits. Thus, the increased inter-subunit distances observed for RGA compared to the other ligands correspond to transiently dissociated receptor dimers, suggesting that quantification by single molecule co-tracking analysis overestimates the fraction of receptor dimers. Importantly, similar re-association in TCZ can also be expected for the other ligands, but, due to the substantially higher molecular complex stability, over substantially longer time-scales that are beyond the time-scale of our single molecule co-tracking experiments. Interestingly though, these transient dissociation events do not seem to compromise signalling as the potency of STAT6 activation by RGA is similar compared to the other ligands. These results suggest that downstream signalling may be maintained during transient dissociation events as schematically depicted in Fig. 8d, which can be rationalized by the very short lifetime of the dissociated state. Such a 'kinetic trapping' of receptor dimers may even be more beneficial considering that under physiologically relevant conditions robust receptor activation is achieved at very low receptor expression levels and sub-stoichiometric occupancies. Kinetic trapping of activated receptor dimers could ensure endocytosis of intact signalling complexes (cf. Fig. 8d), allowing to maintain signalling activity in early endosomes as specifically demonstrated for the IL-4 receptor[25,31]. Thus, rather than being relevant for pre-organization of receptor subunits, our results support the emerging role of plasma membrane microcompartmentalization in sustaining signalling complexes, once they have formed, by passive confinement[47,52].

There is growing evidence that the affinity and dynamics of signalling complexes play a key role for functional selectivity and pleiotropy of cytokines[26,34,62–68]. Here, we could show that the spatiotemporal dynamics of the type II IL-4 receptor signalling complex is governed by an intricate interplay of molecular interactions and receptor confinement in plasma membrane microcompartments, which probably is a generic feature of cytokine receptor dimerization. The early events of receptor dimerization and activation are further convolved by molecular and cellular feedback mechanisms[69] that in turn also critically depend on the molecular interactions involved in receptor assembly. Furthermore, maintaining receptor dimers by TCZs may have important implications for regulating specificity of cellular responses in different cell types that vary in the size and permeability of the MSK meshwork[54]. Taken together, these features yield temporal evolution of downstream signalling, gene expression patterns and cellular decisions that intricately depend on the molecular ligand–receptor interactions as well as the cellular context, thus fundamentally accounting for functional selectivity and pleiotropy of cytokine receptor signalling.

## Methods

**Plasmid constructs.** IL-13 wt, all IL-4 variants and both receptor ectodomains were fused to an N-terminal ybbR-tag (sequence DSLEFIASKLA) for site-specific labelling by phosphopantetheinyl transfer[37]. Coding sequences for IL-4 variants fused to the ybbR-tag were created by side-directed mutagenesis of the IL-4 H59Y sequence that was obtained by gene synthesis, and cloned into the pET28b vector (Novagen) for expression in *Escherichia coli*. Sequences coding for ybbR-tagged IL-13 wt as well as tagless IL-13 wt and IL-4 variants were cloned into the pAcGP67-A vector (BD Biosciences) in frame with an N-terminal gp67 signal sequence and a C-terminal hexahistidine tag for expression in insect cells. For production of IL-4Rα-EC and IL-13Rα1-EC in insect cells, a modified version of the pBAC3 (Novagen) transfer vector was used, comprising a gp64 signal peptide followed by the sequence AMVDSLEFIASKLAGS (which includes the ybbR-tag), the respective receptor ectodomain, a PreScission protease cleavage site and a decahistidine tag at the C-terminus.

pSems HaloTag-IL-13Rα1 and pSems SNAPf-IL-4Rα were generated by inserting IL-13Rα1 and IL-4Rα, respectively, without the N-terminal signal sequences into the pDisplay vector (Invitrogen) via BglII and PstI (IL-13Rα1)/XhoI (IL-4Rα) restriction sites. Subsequently, genes coding for the HaloTag[43] (Promega) and SNAPf-tag[41] (New England Biolabs), respectively, were inserted via the BglII site. The constructs including the Igκ signal sequence of the pDisplay vector were transferred by restriction with EcoRI and XhoI into the pSEMS1-26m vector (Covalys). To obtain the tetracycline inducible reporter plasmid pWHE655-neo TREtight HaloTag-IL-13Rα1 the sequence of the HaloTag-IL-13Rα1 construct including the N-terminal signal sequence of the pDisplay vector was inserted into the pWHE655-neo TREtight vector[53] (kindly provided by Christian Berens, Erlangen) using pSems HaloTag-IL-13Rα1 as template.

IL-13Rα1 and IL-4Rα truncated after the transmembrane domain (IL-13Rα1-TM and IL-4Rα-TM, respectively) were inserted via restriction with BglII and XhoI into the vector pSEMS1-26m including the Igκ signal sequence as well as an N-terminal His₁₀ and a SNAPf-tag[42] to obtain pSems SNAPf-IL-13Rα1-TM and pSems SNAPf-IL-4Rα-TM expression vectors, respectively.

Sequences of all primers used for cloning are provided in Supplementary Table 6.

**Protein production and labelling.** The expression vectors for the ybbR-tagged IL-4 proteins were transformed into *E. coli* Rosetta cells, which were grown at 37 °C and induced by 1 mM IPTG using standard protocols. Cells from 2 l cultures were lysed by sonication and inclusion bodies were isolated by centrifugation. Inclusion bodies were purified by extensive washing steps and unfolded protein was extracted with 6 M guanidium hydrochloride including 5 mM β-mercaptoethanol. Refolding was carried out by dialysis against phosphate buffered saline (PBS; 120 mM NaCl, 2 mM KCl, 3 mM NaH₂PO₄, 7 mM Na₂HPO₄, pH 7.0) using a volume >200-times greater than the sample volume. Refolded IL-4 variants were purified by cation exchange chromatography utilizing High-Performance SP-Sepharose (HiTrap SP HP 5 ml, GE Healthcare) employing a linear NaCl-gradient from 0 M to 1 M at pH 7.0 (20 mM phosphate buffer). Pooled fractions were further purified by SEC on a HiLoad 16/60 Superdex 75 column (GE Healthcare) equilibrated with PBS (pH 7.4).

The ectodomains of IL-13Rα1 and IL-4Rα, all IL-13 proteins as well as tagless IL-4 wt, KFR and RGA were produced in insect cells using a baculovirus expression system. Baculovirus stocks were prepared by transfection and amplification in *Sf*9 cells grown in SF900II media (Invitrogen). High Five (Hi5) suspension cells grown in Insect-Xpress media (Lonza) were infected with the respective baculovirus at an MOI of 1–10. Three to five days after infection, cell supernatants were collected by centrifugation and the proteins were captured by nickel-nitrilotriacetic acid (Ni-NTA) agarose affinity chromatography. Pooled fractions were purified by SEC on a Superdex 200 column (GE Healthcare), and equilibrated in 10 mM 2(4-(2-hydroxyethyl)-1-piperazineethanesulfonic acid (HEPES) (pH 7.2) containing 150 mM NaCl. All protein were purified to >90% homogeneity as confirmed by SDS-PAGE analysis.

Site-specific labelling of ybbR-tagged proteins was carried out by means of Sfp phosphopantetheinyl transferase using self-made CoA-DY647, -Alexa Fluor 568, -Atto 488 or -OregonGreen 488 (Invitrogen) conjugates according to published protocols[70]. Typically, protein concentrations of 10–20 μM reacted. After labelling, the proteins were purified by SEC on a HiLoad 10/300 Superdex 75 column (GE Healthcare). A degree of labelling >90% was obtained as determined by UV/Vis spectroscopy for all ybbR-tagged proteins.

**Interaction kinetics by simultaneous TIRFS-RIf detection.** Simultaneous TIRFS and reflectance interference detection (RIf) were performed using a home-built set-up including a flow system that has been previously described in detail[35,71]. All experiments were performed at 25 °C using HBS buffer (20 mM HEPES, 150 mM NaCl, pH 7.5), which was complemented with 30 mM imidazole to minimize unspecific binding of ligands to free chelator lipids. After mounting a freshly plasma-cleaned RIf transducer into the flow cell, an SSM was generated by injection of small unilamellar vesicles, which were prepared by sonication of 250 μM 1,2-dioleoyl-sn-glycero-3-phosphocholine (DOPC, Avanti Polar Lipids) containing 1 mol% tris-nitrilo triacetic acid-steroyl-octadecylamine (tris-NTA-SOA)[36]. The SSM was then sequentially washed with 500 mM imidazole, 250 mM EDTA and 10 mM NiCl₂. IL-4Rα-EC and IL-13Rα1-EC were injected at concentrations of 10–50 nM to obtain surface concentrations of 1-5 fmol mm⁻². Subsequently, 500 mM decahistidine-tagged maltose binding protein was injected to block excess tris-NTA lipids. After loading the receptors and maltose binding protein to the membrane, 50-100 nM of the respective ^AT488IL-4_D variant or ^OG488IL-13 was injected. Concentrations and injection times used in different types of experiments are provided in the legends of the figures. Dissociation of the fluorescently labelled ligand from the surface was monitored during rinsing for 600 s with HBS at a flow rate of 10 μl s⁻¹. After each experiment, all attached proteins were removed by injecting 500 mM imidazole in HBS to recover the membrane for the next measurement and to check the integrity of the SSM. In these experiments, the mutation H59Y was used for all IL-4 variants to eliminate non-specific binding to the membrane via the complexed Ni(II)-ions observed for IL-4 wt. Importantly, this mutation has no effect on interaction kinetics or activity (cf. Supplementary Fig. 4A,B). Binary interactions were evaluated by a standard pseudo first-order kinetic model using Biaevaluation 3.1 (Biacore). Dissociation from the ternary complex was evaluated by numerical fitting of the differential equations describing a two-step dissociation using Berkeley Madonna 8.3 (UCB, Berkeley, CA, USA). The validity of the utilized model was computationally confirmed by differential equation-based simulations (Supplementary Fig. 22). 2D dissociation rate constants measured by FRET experiments were obtained by fitting an exponential function using OriginPro 8.6 (OriginLab Corp.).

**Reconstitution of receptor subunits in PSM.** SNAPf-IL-13Rα1-TM and SNAPf-IL-4Rα-TM were expressed in HEK 293T cells (DSMZ, ACC 635) transiently transfected at 40% confluency with the respective constructs. A 10 cm dish grown to full confluency for each construct was pelleted after a brief incubation with 50 mM EDTA and after two washing steps with PBS, the cells were homogenized in 500 μl of lysis buffer (25 mM HEPES pH 7.4, 0.5 μl Benzonase, 10 μl protease inhibitors and 20 mM of Triton-X100). After cell lysis, 800 nM of BG-647 or BG-547 (New England Biolabs) was added to the lysis mixture of SNAPf-IL-13Rα1-TM or SNAPf-IL-4Rα-TM, respectively, and the labelling mixture was incubated for 30 min at room temperature. The lysate was then cleared by ultra-centrifugation (55.000g for 45 min at 4 °C). Proteins were extracted from the supernatant by addition of 50 μl of Ni-NTA agarose. After washed and eluted with 60 μl buffer (25 mM HEPES, 300 mM NaCl, 10/500 mM imidazole, 0.6 mM Triton-X100), proteoliposomes were formed by mixing 5 mM of DOPC supplemented with 2 mol% 1,2-dioleoyl-sn-glycero-3-phospho-L-serine (DOPS) in HBS containing 20 mM Triton-X100 with 0.5–1 μl of the eluted protein solution, followed by a detergent extraction by addition of a twofold excess of β-cyclodextrin over the detergent and incubation for 5 min (ref. 72).

PSMs were assembled on glass coverslips, which were coated with diamino-poly(ethylene glycol) and functionalized with palmitic acid[73] and then mounted into an incubation chamber for imaging experiments. Proteoliposomes complemented with DOPC/DOPS liposomes without proteins to a final lipid concentration of 250 μM in HBS were incubated on the coverslip surface for 30 min. After washout of excess vesicles, vesicle rupture was induced by addition of 10% (w/v) PEG8000 in HBS, which was incubated for 15 min. The PSM was then extensively washed with buffer to remove excess lipid material.

Dimerization of reconstituted ^DY547IL-4Rα-TM and ^DY647IL-13Rα1-TM was quantified by dual colour single molecule TIRF microscopy as described below. Ligand concentrations were increased starting from 10 pM up to 50 μM in logarithmically equidistant steps. For each titration, a new PSM was assembled. After equilibration for 5 min, dimerization was quantified by single molecule co-

localization analysis (see below). The dimer fraction was corrected for random co-localizations and for the effective degree of labelling (DOL) of [DY647]IL-13Rα1-TM and [DY547]IL-4Rα-TM, which was determined by co-localization experiments with [DY647]KFR (DOL = 0.95) bound to [DY547]IL-4Rα-TM reconstituted in a PSM.

**Cell culture, transfection and live cell labelling.** HeLa cells (DSMZ, ACC 57) were cultivated at 37 °C and 5% $CO_2$ in MEM with Earle's salts, stable glutamine and phenol red (Biochrom AG, FG0325) supplemented with 10% fetal bovine serum (FBS) (Biochrom AG, S0615), 1% non-essential amino acids (PAA laboratories GmbH M11003) and 1% HEPES (PAA laboratories GmbH) buffer without addition of antibiotics. For microscopy experiments, 20 mm glass coverslips (VWR) were coated with a poly-L-lysine-graft-(polyethylene glycol) (PLL-PEG) copolymer functionalized with RGD[74] to minimize the fluorescent background arising from non-specifically absorbed dye molecules. Cells were plated on the coverslips in 35 mm cell-culture dishes to a density of ∼50% confluence and treated with Penicillin/Streptomycin.

Cells were transfected with plasmids coding for HaloTag-IL-13Rα1 and SNAPf-IL-4Rα using calcium phosphate precipitation as described earlier[75]. Twelve hours after transfection, cells were washed twice with PBS (PAA laboratories GmbH) and fresh medium was added. Transiently transfected cells were typically used for co-tracking and ternary complex quantification experiments 3–4 days after transfection, when the receptor density was close to the endogenous level. Lower expression of HaloTag-IL-13Rα1 was achieved by transient transfection of pWHE655-neo TREtight HaloTag-IL-13Rα1 into HeLa-TET cells that are stably transfected with regulator plasmid pWHE644. Expression was induced by treatment with 10–25 ng ml$^{-1}$ doxycycline for 24 h.

For imaging experiments, coverslips were mounted into custom-made incubation chambers with a volume of 500 µl. HaloTag- and SNAPf-tag-carrying receptor molecules were simultaneously labelled by incubating with 30 nM HaloTag tetramethylrhodamine (TMR) ligand (HTL-TMR, Promega, G8252) and 80 nM of SNAP-Surface647 (BG647, New England Biolabs, S9159S) in culture medium at 37 °C for 15 min. After labelling, cells were washed five times with pre-warmed PBS to remove unreacted dye. For inhibiting actin polymerization cells were treated with 10 µM Latrunculin B for 20 min after labelling at 37 °C. Cholesterol depletion was carried out by treatment with 20 U cholesterol oxidase (ChOx) for 30 min at 37 °C.

**Single molecule TIRF microscopy.** Single molecule imaging experiments were carried out by TIRF microscopy with an inverted microscope (Olympus IX71) equipped with a triple-line total internal reflection (TIR) illumination condenser (Olympus) and a back-illuminated electron multiplying charge coupled device (EMCCD) camera (Ixon DU897D, 512 × 512 pixel, Andor Technology). A × 150 magnification objective with a numerical aperture of 1.45 (UAPO 150 × /1.45 TIRFM, Olympus) was employed for TIR illumination of the samples. DY647-labelled ligands were excited by a 642 nm laser diode (LuxX 642-140, Omicron) at 0.65 mW (power output after passage of the objective). Stacks of 1,000–2,000 frames were recorded with a time resolution of 32 ms per frame (31.25 Hz). For dual-colour acquisition, [DY547]IL-13Rα1 or [TMR]IL-13Rα1 were excited by a 561 nm diode-pumped solid state laser (CL-561-200, CrystaLaser) at 0.95 mW and [DY647]IL-4Rα or DY647-labelled ligands were excited by a 642 nm laser diode (LuxX 642-140, Omicron) at 0.65 mW. Fluorescence was detected using a spectral image splitter (DualView, Optical Insight) that was equipped with a 640 DCXR dichroic beam splitter (Chroma) in combination with bandpass filters for detection of TMR (585/40, Semrock) and of DY647 (690/70, Chroma), projecting each channel onto 512 × 256 pixels (cf. Supplementary Fig. 15A). Stacks of 300–500 images were acquired at 31.25 Hz (32 ms per frame).

Microscopy was performed at room temperature in phenol red-free medium supplemented with an oxygen scavenger and a redox-active photoprotectant consisting of 0.5 mg ml$^{-1}$ glucose oxidase (Roche Applied Science), 0.04 mg ml$^{-1}$ catalase, 4.5 mg ml$^{-1}$ glucose, 1 mM ascorbic acid and 1 mM methyl viologene (all Sigma-Aldrich) to prevent blinking of the DY647 fluorophore and to minimize photobleaching of both dyes[76]. For ligand binding studies, the respective DY647-labelled ligand was added to a final concentration of 2 nM for at least 5 min. After five washing steps with PBS, microscopy medium was added and imaging was started rapidly. Receptor dimerization was probed before and after incubating with the respective ligand at a concentration of 20 nM for at least 5 min if not otherwise stated. Images of the same coverslip were acquired up to 30 min post-stimulation to reduce artefacts that may arise from cellular feedback mechanisms such as receptor endocytosis or stress responses.

**Single molecule analyses.** Single molecule localization and tracking was carried out using the multiple-target tracing algorithm[77] implemented in the customized software package 'SLIMfast' (kindly provided by Christian P. Richter) written in MATLAB (The MathWorks) as described previously in detail[13]. A detection box of 9 × 9 pixels was applied and five deflation loops were performed to ensure the reliable identification of all molecules. A maximum of three nearest neighbours were analysed for recovering particle trajectories and a detection gap between successive frames of three missing points was closed. The local diffusion coefficient was calculated from the past ten frames to model the PDF for observed

displacements and the expected diffusion coefficient was set to an empirically optimized value between 0.09 and 0.15 µm$^2$ s$^{-1}$ (local and expected diffusion equally weighted). With the applied parameter set a sub-micrometer localization precision (∅ ∼ 25 nm) of single molecules was achieved. Immobile molecules (unspecific immobilized dyes and endocytosed labelled receptors) were identified by the 'density-based spatial clustering of applications with noise' (DBSCAN) algorithm[78] as described earlier[79] and removed from the data set.

Channels of dual colour images were aligned by performing a spatial transformation on the basis of a calibration measurement with multicolour fluorescent beads (TetraSpeck microspheres 0.1 µm, Invitrogen) that are visible in both spectral channels. For single molecule co-localization analysis, individual molecules detected in both spectral channels were regarded as co-localized, if the co-particle was found contemporaneously within a distance threshold of 2 pixels (214 nm). In a consecutive step, co-localized particles were subjected to tracking by the multiple-target tracing algorithm to generate co-trajectories. For the receptor distance analysis, only trajectories with a minimum length of 30 steps (∼ 960 ms) were considered.

The diffusion characteristics were analysed by fitting of step-length histograms (increment 32 ms or 1 frame, respectively) that were obtained from a collection of single molecule trajectories of the labelled ligands with a minimum length of 150 frames (4.8 s). The utilized model is based on unhindered Brownian motion using a two-component two-dimensional Gaussian probability distribution function.

PICCS[44] was implemented into a MATLAB script to calculate the mean correlated fraction of localized particles in channel 1 ([TMR]IL-13Rα1) with respect to channel 2 ([DY647]IL-4Rα or DY647-labelled ligand) in a 20–25 µm$^2$ region of interest. Only the first 20 frames of each image stack were evaluated, as the reduction of cross-correlation due to photobleaching was negligible during this short time interval. The mean correlation function $C_{cum}(r)$ was calculated from the particle coordinates with the interval $\Delta r = 0.03$ µm and $r_{max} = 0.7$ µm ($r^2_{max} = 0.49$ µm$^2$). First, the linear part of $C_{cum}$ in the range of $0.1 < r \leq 0.7$ µm was fitted using equation (1)

$$C_{cum}^{lin.}(r) = \rho \cdot \pi r^2 + \alpha \tag{1}$$

to obtain the cross-correlated fraction $\alpha$ and the particle density $\rho$. Keeping these parameters fixed, $C_{cum}$ was fitted over the full range ($0 < r \leq 0.7$ µm) with equation (2)

$$C_{cum}(r) = \alpha \cdot \left(1 - e^{-\frac{r^2}{2\sigma^2}}\right) + \rho \cdot \pi r^2 \tag{2}$$

to obtain the correlation length $\sigma$. The DOL was determined by PICCS analysis of dual-colour images of HeLa cells expressing the type I interferon receptor subunit IFNAR2 fused to both HaloTag and SNAPf-tag (HaloTag-SNAPf-IFNAR2c (ref. 23)) that was labelled with HTL-TMR and BG-647. For HTL-TMR a mean DOL of 32% and for BG-647 of 42% was determined. HaloTag-SNAPf-IFNAR2c was also used as positive control to determine maximum single molecule co-localization and the co-localization precision.

The molecular binding affinity $K_D^T$ of IL-13Rα1 for ligand-bound IL-4Rα and of IL-4Rα for IL-13-bound IL-13Rα1 in the plasma membrane was calculated according to the law of mass action:

$$K_D^T = \frac{([IL-4R\alpha] - \alpha \cdot [IL-13R\alpha1]) \cdot ([IL-13R\alpha1] - \alpha \cdot [IL-13R\alpha1])}{\alpha \cdot [IL-13R\alpha1]} \tag{3}$$

The respective cell surface concentrations of IL-4Rα and IL-13Rα1 were taken from the PICCS analyses ($\rho$, see above) and corrected for the determined DOLs. The correlated fraction $\alpha$ was corrected for the basal of unstimulated cells (caused by random co-localizations and fitting fidelity) and normalized to the maximum dimerization level (62%) that was obtained by setting the correlated fractions of the [DY647]IL-4/[TMR]IL-13Rα1 (∼ 18%) and [DY647]IL-4/[TMR]IL-4Rα (∼ 28%) interactions in relation. The error of the calculated $K_D^T$ values depends on deviations of the DOL, receptor concentrations and uncertainty in PICCS evaluation and was estimated at ± 50%.

**Flow and phospho-flow cytometry.** For quantification of receptor levels in the plasma membrane, HeLa cells were stained at 4 °C with Fluorescein-5-iso-thiocyanate or Phycoerythrin labelled antibodies against IL-13Rα1 (1:100 dilution, R&D cat: FAB1462F), IL-4Rα (1:100, BD Biosciences cat: 552178) and $\gamma_c$ (1:100, Biolegend cat: 338605) for 1 h. Cells were then washed, the levels of surface receptors were measured and the mean fluorescence intensity was compared to an isotype control.

STAT6 phosphorylation after treatment of HeLa cells with different ligands was quantified by phospho-flow cytometry. After ligand incubation cells were fixed and permeabilized with ice-cold methanol (100% v/v). Phosphorylated STAT6 (pSTAT6) was stained with a pTyr641-specific monoclonal antibody coupled to Alexa Fluor 488 (1:50, BD Biosciences cat: 612600). The fluorescence was quantified using an Accuri C6 flow cytometer (BD Biosciences) and background corrected by subtracting the mean fluorescence intensity of an unstimulated sample. Dose–response data were acquired 15 min after induction and the normalized values were plotted against the cytokine concentration to yield dose–response curves. These were fitted to a sigmoidal curve by nonlinear least squares

regression for EC$_{50}$ value calculation. The STAT6 phosphorylation kinetics was quantified at a ligand concentration of 1 μM.

**Model simulations.** A simple spatial diffusion-reaction model (DAIDS) was implemented in MATLAB (Version R2015a) that includes a submicroscopic, semipermeable meshwork to mimic the MSK. The simulation with time-resolution $\Delta t_{sim}$ consists of two massless molecular species (1: binary complex of ligand and high affinity ligand-binding receptor subunit, 2: accessory receptor subunit) with the corresponding particle surface concentrations $N_1$ and $N_2$. The particles are diffusing with diffusion constant $D$ according to a simple isotropic random walk model[80] in a two-dimensional space (plasma membrane), having the dimensions $8.99 \times 8.99$ μm ($80.78$ μm$^2$) or $84 \times 84$ pixels (pixel-size 0.107 μm, taken from experimental setup), respectively. The membrane is divided into right triangular compartments with leg side lengths of 74.9 nm (0.7 pixel) that are arranged in an alternating order (cf. Supplementary Fig. 20). Upon collision with the compartment boundaries, particles are allowed to pass with a probability $p_{hop}$. If each of a molecule of species 1 and 2 are getting closer together than a defined association threshold $d_a$, a ternary complex is formed with probability $p_a$. The dissociation of ternary complexes occurs with probability $p_d$. Further details of the simulations are provided in the Supplementary Methods.

**Code availability.** Full, ready-to-use DAIDS code is available via figshare at https://doi.org/10.6084/m9.figshare.4868180.v1. The 'SLIMfast' software package for single molecule data evaluation is available on request from Christian P. Richter (christian.richter@biologie.uni-osnabrueck.de).

**Data availability.** The data that support the findings of this study are available from the authors on reasonable request (see author contributions for specific data sets).

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

## Acknowledgements

We thank Dr C. Berens for providing the Tet-inducible expression system, Dr R. Kurre for support with fluorescence microscopy, C. Reynolds and G. Hikade for excellent experimental support, C. P. Richter for providing single molecule localization and tracking tools and the members of the Piehler, Garcia and Müller laboratories for helpful advice and discussion. This project was supported by the SFB 944 (P8 and Z) from the Deutsche Forschungsgemeinschaft (to J.P.), NIH-RO1AI51321 and HHMI (to K.C.G.) as well as by the BMBF project IlReMu (to T.D.M.).

## Author contributions

D.R., T.D.M. and J.P. conceived the project; M.S., M.K., I.M., H.K. and T.D.M. produced and labelled interleukins and receptor ectodomains; D.R., O.Be., D.P., M.G. and J.P. designed and performed TIRFS-RIf studies; D.R., H.W., S.W., and J.P. designed and performed single molecule microscopy experiments; I.M. performed flow and phospho-flow cytometry experiments; O.Bi. performed PSM experiments; D.R., P.S. and T.S. implemented data evaluation by PICCS; D.R. developed and performed MATLAB simulation studies; J.P. and D.R. wrote the manuscript with the assistance of I.M., K.C.G. and T.D.M. The final version of the manuscript was read and approved by all authors.

## Additional information

**Competing interests:** The authors declare no competing financial interests.

