## [Peer Review File · Nature Communications]

Reviewers' comments:

Reviewer #1 (Remarks to the Author):

The authors analyze molecular and cellular determinants of cytokine receptor assembly. They use receptors reconstituted in artificial membranes and single molecule localization microscopy in the membrane of live cells. Ligand induced dimerization was independent of receptor binding affinity. The authors additionally found that actin-dependent membrane microdomains play a role in receptor dimer stabilization.

This is a very interesting study with important novel implications for cytokine biology. There are some points the authors should consider.

Major points:

1. The authors should state how they checked the quality of the ligand- and receptor-proteins used in this study. Data could be shown as supplemental material.
2. A very important point of the paper is the role of actin-dependent membrane microcompartmentation. The authors use a single dose (10 μ M) of the drug Lantrunculin B to partially inhibit actin polymerization. Many studies use different doses of this drug. Can the authors give a rationale?
3. The authors should present pictures of cells with actin staining treated with different doses of Lantrunculin B to demonstrate the effect of the drug.
4. Along the same line: what is the molecular mechanism of Lantrunculin B? Are there other drugs with a similar effect, which could be used to corroborate this important finding?
5. Along the same line: can the authors show a dose-response effect of Lantrunculin B?
6. The somehow related reports by Becker et al (Science 2010) and (Science 2014) could be mentioned and discussed.

Minor points:

1. On p4, the authors refer to the affinity of IL-4 to IL-4Ra, which is lower than 1 nM, as 'very high affinity'. This is questionable since in some cytokine systems such an affinity is considered a low affinity. The term 'very high affinity' should be avoided.
2. The tagging procedure (ybbR-tag) should be shortly explained for the non-specialized reader (p5).
3. The tagging procedure (SNAPf-tag) should be shortly explained for the non-specialized reader (p6).
4. The tagging procedure (Halo-tag) should be shortly explained for the non-specialized reader (p9).
5. Particle image cross-correlation (PICCS) analysis should be shortly explained for the non-specialized reader (p10).
6. If possible, a newer review instead of ref. 1 should be cited.
7. Ref. 8 is incomplete
8. Ref. 64 is incomplete

Reviewer #2 (Remarks to the Author):

Richter et al studied the ligand-induced interaction of Interleukin-4 and Interleukin-13 receptors in vitro and in living cells. Using different mutants, they found up to 100-fold reduced affinity in the in vitro experiments, which did not show up at the plasma membrane. The discrepancy is attributed to the membrane skeleton, since it increases the apparent colocalization time. This is a highly comprehensive study, with numerous data supporting the proposed model. The outcome is interesting for a broad readership and in principle would warrant publication in Nature

Communications.

A few major and minor points, however, need to be addressed.

Major

1.) The key point of the paper is the difference in interaction parameters between IL-4/IL-4R α \leftrightarrow IL-13R α 1 and RGA/IL-4R α \leftrightarrow IL-13R α 1 in vitro, while similar results were found in live cells. The two measurements, however, were done differently: in vitro experiments were performed by analysis of the unbinding of fluorescent ligand presumably from the monomeric form of the receptor, whereas live cell data were recorded by single molecule microscopy of the fluorescently labelled receptors. I strongly encourage the authors to show that the data obtained from the in vitro approach quantitatively agree with the live cell data, by performing on the very same sample both approaches.

2.) The in vitro data were obtained using rather indirect measurements, hence the fits used to extract the in vitro binding parameters need to be validated. How stable are the results against changes of the assumed binding model? For example, it is assumed that the ligand unbinds exclusively from the monomer; what would be the consequences of ligand unbinding from dimers, say, with reduced off-rate? As stated above, it would be better if the dissociation of the receptors could be directly quantified, as done in the live cell experiments.

3.) I don't see how the hop diffusion model explains the differential behavior of RGA/IL-4R α \leftrightarrow IL-13R α 1 versus IL-4/IL-4R α \leftrightarrow IL-13R α 1. Wouldn't the membrane skeleton also increase the rebinding likelihood for IL-4/IL-4R α \leftrightarrow IL-13R α 1?

4.) I don't understand the biological significance of the model. On the one hand, as demonstrated by the authors the presence of a membrane skeleton leads to increased rebinding by keeping the two partners within the same compartment. On the other hand, however, I would expect reduced on-rates in general, since receptors have to overcome the barrier between compartments in order to reassociate. The overall K_d depends only on binding energies and should be identical. Hence, the hop diffusion model shouldn't lead to enhanced dimerization per se.

Minor

1.) Page 4, "The IL-13/IL-13R α 1 complex, however, binds to IL-4R α with a binding affinity of 10-20 nM". These should be 2D affinities.

2.) Page 5: "Neither the ybbR-tag nor the H59Y mutation affected receptor binding properties or activity of IL-4". How do you know?

3.) Page 5: "Importantly, the interaction with IL-13R α 1 is not affected by this mutation as the corresponding binding interface is unaltered". Again, how do you know? There could be allosteric effects.

4.) Supplementary Fig. 2: The axis of the insets are too small to read.

5.) Page 5/6, "Dissociation of AT488IL-4D was substantially slower compared to experiments with only IL-4R α -EC due to ternary complex formation (Figure 1C)": It would be clearer for comparison to include the respective curve in Fig. 1B.

6.) Page 9, "revealed a significantly reduced mobility of DY647KFR compared to DY647DN". How was the significance inferred? The difference doesn't seem too strong.

7.) Page 11, "Interestingly though, a significantly increased correlation length was observed for DY647IL-4R α and TMRIL-13R α 1 upon stimulation with RGA". What is the reason for the large error bars in Fig. 4E for RGA?

8.) Page 12, "the length of co-trajectories was substantially reduced for RGA, but not for KFR". Could this also be due to the differentially enhanced mobilities?

Reviewer #3 (Remarks to the Author):

This paper aims at deciphering the assembly of the type II IL4R. The study is divided in two parts. The first one aims at quantifying the affinity and stability of the complex reconstituted in artificial membrane. Next, the authors used single molecule tracking to analyze the complex in living cells and under different physiological conditions. However, the comparison of the results observed in reconstituted system and at the plasma membrane of HeLa cells revealed significant differences on the efficacy of the ternary complex formation.

Among the main conclusions, the authors reported that (1) the predimerization of IL4R/IL13R are not detected at the plasma membrane of living cells, (2) the IL4R/IL13R dimerization was not exclusively controlled by the binding affinities of the ligands, and (3) the compartmentation of the plasma membrane by cortical actin as a key element regulating the stabilization of the ternary complex. The later was sustained by modeling hop diffusion.

In the whole, this work is of general interest for a broad cell biologist community and to my understanding, the experiments are well conducted with appropriate controls and robust analytical tools. Still, the following points need to be clarified or modified:

(1) At the experimental and conceptual levels. Actually, the authors focused their interpretation on the difference between the in vitro and in vivo systems on the stabilization function of the actin-dependent membrane compartmentation.

This does not take into account the possible role of the plasma membrane as facilitator of the ternary complex formation as compared to the in vitro system where the lipid bilayer has a passive role of support. To clarify that specific point, the authors should also explore to what extent the lipid composition of the plasma membrane might contribute or not to accelerate/facilitate this complex formation. Indeed, the two mechanisms are not exclusive but can be connected each other. As such, I would strongly suggest to focus on in vivo experiments by SPT under pharmacological or enzymatic conditions modifying the lipid composition of the plasma membrane in living cells.

(2) Moreover, the current version of the manuscript need to be modified to make the nomenclature used to identify the molecular components more immediately explicit for both the in vitro and in vivo biological models. I would suggest to modify and to append the table currently in the "supplementary methods" section (page 2) within the core of the manuscript.

(3) To my understanding, the word "meshwork" instead of "microdomains" more accurately refers to the actin-based membrane confinement.

Reviewer 1:

Major points

1. *The authors should state how they checked the quality of the ligand- and receptor-proteins used in this study. Data could be shown as supplemental material.*

All proteins were purified by size exclusion chromatography (SEC) to ensure folding into globular, monomeric proteins and their purity was confirmed by SDS PAGE. In the revised manuscript, we have described protein purification and quality control in more detail and included SEC and SDS PAGE data in the new Supplementary Fig. 1 and 2, respectively. Moreover, functional integrity and activity of labeled and unlabeled proteins was confirmed by SPR or TIRFS-Rlf detection and phospho-flow cytometry. These data were included into the new Supplementary Fig. 3.

2. *A very important point of the paper is the role of actin-dependent membrane microcompartmentation. The authors use a single dose (10 μ M) of the drug Latrunculin B to partially inhibit actin polymerization. Many studies use different doses of this drug. Can the authors give a rationale?*

Destabilizing the cortical actin skeleton by drugs interfering with actin polymerization is rather delicate and has been studied in detail by the Kusumi and the Marguet groups, who are renowned experts in this field. We have consulted thoroughly with both groups to ensure optimum conditions in these experiments. They consistently recommended using Latrunculin A or B, which bind to monomeric G-actin and prevent polymerization into F-actin. Short-term effects of Latrunculin B are similar to those of Latrunculin A, but Latrunculin A is slightly less potent (Spector, I. et al. *Cell Motil Cytoskeleton*, 13, 127-144, 1989). In contrast to Latrunculin A/B, Cytochalasin D is, probably due to compensatory effects, not as suitable to interfere with MSK-dependent plasma membrane microcompartmentation, as previously demonstrated by the Kusumi group (*Biophysical Journal*, Volume 95, pp. 435-450, 2008). The LatB concentration applied in our study was fine-tuned to affect the permeability of the MSK while maintaining cell viability and attachment to the coverslip surface via focal adhesion, which is crucial both for microscopy and for functional assays. Images of the cytoskeleton before and after treatment with 10 μ M LatB provided in the revised manuscript (new. Supplementary Fig. 17, s. below) highlight that mainly the cortical actin skeleton is affected. Under these conditions, a significant increase in receptor mobility was observed, which is a key prerequisite for testing our hypothesis that confinement by the MSK is responsible for the surprising robustness of receptor dimerization in the plasma membrane. At lower LatB concentrations, changes in MSK and receptor diffusion constants are insignificant. Therefore we decided to stick to a single, optimized concentration of LatB, which could be reliably applied for imaging and functional studies. We have included some of these considerations into the revised manuscript (Line 338-343).

3. *The authors should present pictures of cells with actin staining treated with different doses of Latrunculin B to demonstrate the effect of the drug.*

In the revised manuscript, we have included a representative live cell fluorescence micrograph of a HeLa cell expressing Lifeact-mEGFP before and after treatment with 10 μ M LatB (new

Supplementary Fig. 17). These images show significant changes in the fine-structure of actin filaments, while the stress fibers are maintained. The corresponding DIC images reveal characteristic formation of membrane blebs corroborating the loss in stability of the cortical cytoskeleton.

4. *Along the same line: what is the molecular mechanism of Lantrunculin B? Are there other drugs with a similar effect, which could be used to corroborate this important finding?*
See answers to point (2) and (3).

5. *Along the same line: can the authors show a dose-response effect of Lantrunculin B?*
See answers to point (2) and (3).

6. *The somehow related reports by Becker et al (Science 2010) and (Science 2014) could be mentioned and discussed.*

We have included Becker et al. (Science 2010) into the revised manuscript (line 455-457), but we were not able to identify the second paper the reviewer referred to.

Minor points

1. *On p4, the authors refer to the affinity of IL-4 to IL-4Ra, which is lower than 1 nM, as 'very high affinity'. This is questionable since in some cytokine systems such an affinity is considered a low affinity. The term 'very high affinity' should be avoided.*

We agree to the reviewer and therefore we have changed these ill-defined terms into more precise statements (e.g. "sub-nanomolar", line 66/67).

2. *The tagging procedure (ybbR-tag) should be shortly explained for the non-specialized reader (p5).*

We have included more details on the ybbR-tag labeling into the Results section (line 96-99) and into the Methods section.

3. *The tagging procedure (SNAPf-tag) should be shortly explained for the non-specialized reader (p6).*

We have included more details on the SNAPf-tag labeling into the Results section (line 242-245).

4. *The tagging procedure (Halo-tag) should be shortly explained for the non-specialized reader (p9).*

We have included more details on the HaloTag labeling into the Results section (line 242-245).

5. *Particle image cross-correlation (PICCS) analysis should be shortly explained for the non-specialized reader (p10).*

We have included more explanations on the PICCS evaluation into the Results section (line 263-270). Further information is provided as Supplementary Fig. 15C.

6. *If possible, a newer review instead of ref. 1 should be cited.*

There are very few general reviews on cytokine receptor therapeutics, but rather focused on specific systems. We have therefore included another review on the IL-4/IL-13 system.

7. *Ref. 8 is incomplete*

References were thoroughly updated and completed.

8. *Ref.64 is incomplete*

References were thoroughly updated and completed.

Reviewer 2:

Major points

1. *The key point of the paper is the difference in interaction parameters between IL-4/IL-4R α \leftrightarrow IL-13R α 1 and RGA/IL-4R α \leftrightarrow IL-13R α 1 in vitro, while similar results were found in live cells. The two measurements, however, were done differently: in vitro experiments were performed by analysis of the unbinding of fluorescent ligand presumably from the monomeric form of the receptor, whereas live cell data were recorded by single molecule microscopy of the fluorescently labelled receptors. I strongly encourage the authors to show that the data obtained from the in vitro approach quantitatively agree with the live cell data, by performing on the very same sample both approaches.*
2. *The in vitro data were obtained using rather indirect measurements, hence the fits used to extract the in vitro binding parameters need to be validated. How stable are the results against changes of the assumed binding model? For example, it is assumed that the ligand unbinds exclusively from the monomer; what would be the consequences of ligand unbinding from dimers, say, with reduced off-rate? As stated above, it would be better if the dissociation of the receptors could be directly quantified, as done in the live cell experiments.*

We agree with the reviewer that these two points are important and therefore we have substantially extended the in *vitro* part to meet these justified concerns:

(i) We have included control experiments validating the model used for fitting ligand dissociation from the ternary complex. TIRFS-RIf control experiments with different excess of IL-13R α 1 over IL-4R α reveal spontaneous dissociation of IL-4 from ternary complex to be > 200-fold slower ($k \sim 0.0001 \text{ s}^{-1}$) than from binary complex. This data nicely supports our model and was included into the main manuscript (new Fig. 1C). We have included additional modeling data for the reviewer (s. appendix below) confirming that direct dissociation from the ternary complex can be neglected (error < 10%) for the evaluation under these conditions.

(II) However, we have also more extensively explored dimerization of transmembrane IL-4R α and IL-13R α 1 reconstituted into PSM. These data suggest that dimerization of the transmembrane receptor is substantially enhanced compared to the tethered receptor subunits, while the strong differences in dimerization efficiency of the IL-4 mutants is nicely corroborated by these experiments. We have included this data as separate section (new Fig. 3) into the revised manuscript (line 175-202).

3. *I don't see how the hop diffusion model explains the differential behavior of RGA/IL-4R α ↔ IL-13R α 1 versus IL-4/IL-4R α ↔ IL-13R α 1. Wouldn't the membrane skeleton also increase the rebinding likelihood for IL-4/IL-4R α ↔ IL-13R α 1?*

We fully agree with the reviewer that the same effect is expected for all IL-4 mutants, but it is more clearly visible for RGA because of the time scale and the time resolution of our experiments: Since the dimer lifetime of wt and KFR complexes is >1 s, dissociation events during the time of the measurement are much more rare than for RGA. Indeed, the distance distributions shown in Fig. 6F also contain a small fraction of larger distances for the other agonists (Supplementary Table 5). We have explained this data and our interpretations in the revised manuscript (line 384-388 and line 435-441).

4. *I don't understand the biological significance of the model. On the one hand, as demonstrated by the authors the presence of a membrane skeleton leads to increased rebinding by keeping the two partners within the same compartment. On the other hand, however, I would expect reduced on-rates in general, since receptors have to overcome the barrier between compartments in order to associate. The overall K_d depends only on binding energies and should be identical. Hence, the hop diffusion model shouldn't lead to enhanced dimerization per se.*

This is an important point, which we actually did not properly raise. Indeed, the equilibrium cannot be altered by passive diffusion confinement. Also, the increased inter-receptor distance observed for RGA suggests that the complex is dissociated for some time, but then re-associates and continues diffusion as a dimer. Under these conditions, the co-tracking analysis does not account for such transient dissociation events and therefore does not properly reflect the equilibrium, but overestimates the dimerization levels. But it seems that such transient dissociation events do not severely affect signaling activity as the maximum pSTAT6 activation is not affected by the RGA-mutations. However, in the case that the dimers are not properly confined to ensure rapid re-association (i.e. increased hopping probability in presence of LatB), we see a loss in pSTAT6 activation. We have included these important mechanistic and functional consequences into the Discussion sections (line 433-441, respectively). We also changed the title of the paper avoid the misleading term “stabilize”.

Minor points

1. *Page 4, “The IL-13/IL-13R α 1 complex, however, binds to IL-4R α with a binding affinity of 10-20 nM”. These should be 2D affinities.*

This was mistaken and corrected in “IL-13R α 1-EC”.

2. *Page 5: “Neither the ybbR-tag nor the H59Y mutation affected receptor binding properties or activity of IL-4”. How do you know?*
3. *Page 5: “Importantly, the interaction with IL-13R α 1 is not affected by this mutation as the corresponding binding interface is unaltered”. Again, how do you know? There could be allosteric effects.*

We have provided additional binding and activity data into the revised manuscript to confirm that the ybbR-tag and the H59Y mutation neither affect binding nor activity in new Supplementary Fig. 4 and Supplementary Table 3. These include representative binding curves of IL-4R α -

ECD to surface-tethered ybbR-IL-4 H59Y (Supplementary Fig. 4A), binding data for all mutants (Supplementary Table 3) and the comparison of dose response pSTAT activation curves for IL-4 WT, ybbR-IL-4 H59Y, IL-4 RGA, ybbR-IL-4 H59Y RGA, IL-4 KFR, IL-4 H59Y KFR (Supplementary Fig. 4B). Moreover, we present binding curves for the interaction of H59Y KFR and H59Y K84D KFR, respectively, in complex with soluble IL-4R α -ECD, with immobilized IL-13R α 1-EC (Supplementary Fig. 4C), Likewise dimerization efficiency in PSM is not changed for the IL-4 K84D mutant compared to IL-4 (new Figure 3), demonstrating K84D does not affect the interaction with IL-13R α 1.

4. *Supplementary Fig. 2: The axis of the insets are too small to read.*

We have updated this Figure to ensure readability.

5. *Page 5/6, "Dissociation of AT488IL-4D was substantially slower compared to experiments with only IL-4R α -EC due to ternary complex formation (Figure 1C)": It would be clearer for comparison to include the respective curve in Fig. 1B.*

Since we also performed measurements with different IL-13R α 1-EC surface concentrations we have included a new Figure 1C to show the direct comparison of binary and ternary complex formation for ^{AT488}IL-4_D.

6. *Page 9, "revealed a significantly reduced mobility of DY647KFR compared to DY647DN". How was the significance inferred? The difference doesn't seem too strong.*

The differences are small, but also the standard deviations are small (1-2%, s. Table 2). Additionally, we have determined the statistical significance by comparing the diffusion constants from each cell (new Supplementary Fig. 14C). Student's t-test analysis of these data yields very high significance of the difference ($p < 0.0001$).

7. *Page 11, "Interestingly though, a significantly increased correlation length was observed for DY647IL-4R α and TMRIL-13R α 1 upon stimulation with RGA". What is the reason for the large error bars in Fig. 4E for RGA?*

This is an important point, because the large error bar of RGA (as well as the broad distribution in Fig. 6F) nicely supports that both small distances (within ternary complexes) and larger distances (for dissociated complexes) occur. We have pointed this out in the revised manuscript (line 308-314).

8. *Page 12, "the length of co-trajectories was substantially reduced for RGA, but not for KFR". Could this also be due to the differentially enhanced mobilities?*

We have provided the data that confirm identical mobility of KFR- and RGA-induced dimers in presence of LatB (new Supplementary Fig. 19).

Reviewer 3:

1. *At the experimental and conceptual levels. Actually, the authors focused their interpretation on the difference between the in vitro and in vivo systems on the stabilization function of the actin-*

dependent membrane compartmentation. This does not take into account the possible role of the plasma membrane as facilitator of the ternary complex formation as compared to the in vitro system where the lipid bilayer has a passive role of support. To clarify that specific point, the authors should also explore to what extent the lipid composition of the plasma membrane might contribute or not to accelerate/facilitate this complex formation. Indeed, the two mechanisms are not exclusive but can be connected each other. As such, I would strongly suggest to focus on in vivo experiments by SPT under pharmacological or enzymatic conditions modifying the lipid composition of the plasma membrane in living cells.

We fully agree that lipids could contribute to enhanced receptor dimerization. Here, we focused on the MSK as the fundamental basis of plasma membrane organization. While detailed analyses of lipid-based effects are beyond the scope of this work, we have, in the revised version, included data on how enzymatic cholesterol depletion affects the diffusion of the endogenous receptor (new Supplementary Fig. 18). We find, with high significance ($p < 0.0001$, Student's t-test), that cholesterol depletion does not change the diffusion of cell surface receptor-bound KFR and DN, respectively. These new data were included into the Results and the Discussion sections (line 341-344 and line 424-425).

2. *Moreover, the current version of the manuscript need to be modified to make the nomenclature used to identify the molecular components more immediately explicit for both the in vitro and in vivo biological models. I would suggest to modify and to append the table currently in the "supplementary methods" section (page 2) within the core of the manuscript.*

We have left the Table in the SI, but leave the decision to the editor. By re-sorting the results sections into SSM and PSM experiments and by using a stringent nomenclature, we tried to improve readability.

3. *To my understanding, the word "meshwork" instead of "microdomains" more accurately refers to the actin-based membrane confinement.*

We only demonstrate the relevance of the MSK, but we cannot entirely exclude that other factors such as lipids contribute. We therefore now consistently use the terms "microcompartmentation" and "transient confinement zone" (TCZ) instead, and correspondingly also changed the title.

Appendix

Experimental data ^{AT488}IL-4_D dissociation

$$k_d^B / \text{app. } k_d^* > 200$$

Simulations — current model $k_d^* = 0$ $k_a^T = 3.8 \cdot 10^{15}$

k_d^*	0	1/10	1/20	1/50	1/100	1/200	$\times k_d^B$
k_a^T	3,8	77,5	8,7	5,0	4,3	4,0	$\times 10^{15}$
DEV	-	20,40	2,30	1,32	1,14	1,07	-fold
	-	2040	230	132	114	107	%

Reviewers' Comments:

Reviewer #1:

Remarks to the Author:

The authors analyze molecular and cellular determinants of cytokine receptor assembly. They use receptors reconstituted in artificial membranes and single molecule localization microscopy in the membrane of live cells. Ligand induced dimerization was independent of receptor binding affinity. The authors additionally found that actin-dependent membrane microdomains play a role in receptor dimer stabilization.

This is a very interesting study with important novel implications for cytokine biology.

The authors have addressed all questions and concerns of this reviewer in a satisfactory fashion.

Reviewer #2:

Remarks to the Author:

All points were addressed. I suggest adding the appendix from the rebuttle letter to the paper.

Reviewer #3:

Remarks to the Author:

The authors satisfactory address the clarifications/modifications I asked